# ASPSCR1-TFE3 reprograms transcription by organizing enhancer loops around hexameric VCP/p97

Amir Pozner[1,2,3,14], Li Li[1,2,3,14], Shiv Prakash Verma[1,2,3,14], Shuxin Wang[4], Jared J. Barrott[1,2,3], Mary L. Nelson [1,2,3], Jamie S. E. Yu[5], Gian Luca Negri [6], Shane Colborne[6], Christopher S. Hughes [6], Ju-Fen Zhu[1,2,3], Sydney L. Lambert[1,2,3], Lara S. Carroll[1,2,3], Kyllie Smith-Fry[1,2,3], Michael G. Stewart [1,2,3,4], Sarmishta Kannan[1,2,3], Bodrie Jensen[1,2,3], Cini M. John [7], Saif Sikdar [7], Hongrui Liu[7], Ngoc Ha Dang[8], Jennifer Bourdage[8], Jinxiu Li[1,2,3], Jeffery M. Vahrenkamp[2,3], Katelyn L. Mortenson [2,3], John S. Groundland[1,3], Rosanna Wustrack[9], Donna L. Senger [8,10], Franz J. Zemp[9], Douglas J. Mahoney [9], Jason Gertz[2,3], Xiaoyang Zhang [2,3], Alexander J. Lazar [11], Martin Hirst [6,12], Gregg B. Morin [6,13], Torsten O. Nielsen [5], Peter S. Shen [4] & Kevin B. Jones [1,2,3] ✉

The t(X,17) chromosomal translocation, generating the ASPSCR1::TFE3 fusion oncoprotein, is the singular genetic driver of alveolar soft part sarcoma (ASPS) and some Xp11-rearranged renal cell carcinomas (RCCs), frustrating efforts to identify therapeutic targets for these rare cancers. Here, proteomic analysis identifies VCP/p97, an AAA+ ATPase with known segregase function, as strongly enriched in co-immunoprecipitated nuclear complexes with ASPSCR1::TFE3. We demonstrate that VCP is a likely obligate co-factor of ASPSCR1::TFE3, one of the only such fusion oncoprotein co-factors identified in cancer biology. Specifically, VCP co-distributes with ASPSCR1::TFE3 across chromatin in association with enhancers genome-wide. VCP presence, its hexameric assembly, and its enzymatic function orchestrate the oncogenic transcriptional signature of ASPSCR1::TFE3, by facilitating assembly of higher-order chromatin conformation structures demonstrated by HiChIP. Finally, ASPSCR1::TFE3 and VCP demonstrate co-dependence for cancer cell proliferation and tumorigenesis in vitro and in ASPS and RCC mouse models, underscoring VCP's potential as a novel therapeutic target.

In some cancers, simple initiating genetic changes impact genome-wide transcription to drive transformation. Alveolar soft part sarcoma (ASPS), a relentless soft tissue malignancy with a predilection for adolescents and young adults, frequently demonstrates a t(x;17) chromosomal translocation as the singular genetic alteration[1]. The resultant *ASPSCR1::TFE3* fusion gene (hereafter abbreviated *AT3*)

defines ASPS as well as some Xp11 renal cell carcinomas (RCCs)[2–4]. Conditional AT3 expression drives sarcomagenesis in perivascular tissues of the brain and retina in the mouse, faithfully recapitulating human ASPS histology and transcriptome[5].

AT3 is not readily targetable, like most of the chromatin-related fusion oncoproteins that drive their respective cancers, given their

large intrinsically disordered regions and lack of enzymatic sites for the binding of small molecules[6]. The search for co-factors to each fusion oncoprotein therefore has urgency and impact. As a working definition, we defined a co-factor for a transcription factor fusion oncoprotein as a second protein that interacts directly with the oncoprotein, interacts at the site of function for that oncoprotein (at specific binding sites on chromatin), and enables the biological function of the fusion oncoprotein (the transcriptional impact of the fusion oncoprotein depends on the presence of the second protein.) There are two types of AT3 fusions, with the activation sequence from the third exon of TFE3 included in type 2 (AT3.2 herein) and omitted from type 1 (AT3.1). To both types, TFE3 contributes its DNA-binding domain, with imputed function in the fusion[7]. However, native ASPSCR1 is not generally present in the nucleus and lends no likely chromatin modifying or transcription-regulating function[8]. Native ASPSCR1 in the cytoplasm antagonizes valosin containing protein (VCP/p97/Cdc48)[9], an AAA+ type ATPase, that otherwise assembles into homo-hexameric rings to segregate, unfold, or extract proteins from membranes or complexes[10–12]. ASPSCR1-mediated disassembly of VCP hexamers[13,14] regulates their role in autophagy, quality control of protein folding in the endoplasmic reticulum, and shuttling of ubiquitinated substrates to the proteasome[15,16].

Here, we identify VCP as a co-factor to AT3, beginning with comparative proteomics of sarcomas from our mouse genetic model of ASPS as well as human cancer cell lines that natively express AT3. We characterize the multi-valent nature of the AT3:VCP interaction, with VCP retaining its hexameric structure and therefore its enzymatic activity. We then interrogate the role the AT3:VCP interaction plays in transcriptional regulation, enhancer loop formation, and oncogenesis.

## Results

### AT3 interacts with VCP in alveolar soft part sarcoma and renal cell carcinoma cells

Immunoprecipitation (IP) for AT3 or control IgG in nuclear extracts from FU-UR-1 cells (human RCC cell line driven by AT3.2) and mouse ASPS tumors demonstrated similar interaction profiles between the species by mass spectrometry-based proteomics, with VCP, MATR3, and MTA2 showing the strongest protein associations (Fig. 1a, Supplementary Fig. 1a). These associations were confirmed by IP western blots (WBs) from FU-UR-1 and ASPS-1 cells (AT3.1-expressing human cancer cell line), as well as mouse ASPS tumors, but not controls (Fig. 1b, c).

To identify the specific interacting regions of AT3, 3XFLAG-tagged constructs were transfected into HEK293T cells for FLAG-IP. Tagged AT3, full-length ASPSCR1, and truncated ASPSCR1 (trASPSCR1, the amino-terminal 311 amino acids in AT3) each robustly co-IPed VCP (Fig. 1d, e, Supplementary Fig. 1b). MATR3 and MTA2 interacted with the TFE3 portion of AT3; MTA2 interacted with full-length ASPSCR1; however, neither interacted with the ASPSCR1 portion of AT3 (Fig. 1e). We therefore focused further experiments on the AT3:VCP interaction.

### AT3 interacts with VCP in the nucleus of cancer cells

Strong nuclear TFE3 immunohistochemistry confirms AT3 expression clinically in RCC or ASPS tissues[17]. VCP varies in nuclear-to-cytoplasmic abundance[18]. Immunofluorescence demonstrated nuclear TFE3 (as AT3) and strong nuclear VCP in four frozen human ASPS specimens (Fig. 1f, Supplementary Fig. 1c). Controls (two Ewing sarcomas, two clear cell sarcomas) had no TFE3 and scant nuclear VCP immunofluorescence. Immunohistochemistry also identified significant nuclear VCP in both mouse and human ASPS tumor tissue microarrays (Supplementary Fig. 1d). Although VCP has nuclear presence in some cells, its strong nuclear presence in AT3-expressing cells supports the AT3:VCP interaction.

HEK293T cells have very little nuclear VCP. HEK293T cells were co-transfected to express a VCP-GFP fusion and either control red

fluorescent protein (mRFP) or an AT3-mRFP fusion. Fluorescent images and fractional WBs demonstrated that cytoplasmic VCP-GFP in controls shifted toward nuclear VCP-GFP with AT3-mRFP (Fig. 1g). Proximity ligation assay (PLA) in human tissue sections identified TFE3 and VCP proximity (<40 nm) only in nuclei of ASPSs, not controls (Fig. 1h, i, Supplementary Fig. 1e–g).

### AT3 interacts with hexameric assemblies of VCP

Full-length ASPSCR1 disassembles VCP hexamers through C-terminal motifs (residues 313–553) not included in AT3[14]. In contrast, N-terminal constructs of ASPSCR1 (residues 1–279) interact with, but not to disassemble VCP hexamers[8].

Overexpressed FLAG-tagged ASPSCR1 in HEK293T cells disassembled VCP hexamers, by blue native polyacrylamide gel electrophoresis (BN-PAGE) and WB for VCP; AT3 did not (Fig. 2a). FLAG-IPs identified VCP as the most prominent interacting protein with either AT3 or ASPSCR1 by mass spectrometry (Fig. 2b). Negative staining and transmission electron microscopy (TEM) of coIPs revealed hexameric VCP with AT3, but not ASPSCR1 (Fig. 2c, d).

To further investigate VCP-hexamer dynamics, purified recombinant VCP was mixed with equimolar ASPSCR1 or AT3 in vitro. In size exclusion chromatography, VCP:ASPSCR1 complexes eluted later than VCP alone, consistent with known ASPSCR1-mediated disassembly of VCP hexamers (Supplementary Fig. 2a, b). In contrast, VCP-AT3 complexes eluted earlier, suggesting that hexamers remained intact. Negative stain TEM revealed typical hexameric VCP assemblies with AT3 but not ASPSCR1 after in vitro mixing (Supplementary Fig. 2c, d).

Next, we utilized a carboxy-half portion of ASPSCR1, excluded from AT3 (CΔ, Fig. 2e), previously reported to disassemble VCP hexamers[14]. Overexpressed in HEK293T cells, CΔ disassembled VCP hexamers by VCP-WB after BN-PAGE (Fig. 2f). Co-transfection of CΔ with AT3 reduced the presence of higher molecular weight assemblies in FLAG-AT3-IP (Fig. 2g). FLAG-AT3-IP without co-transfected CΔ recovered assemblies as large as 1050 kD, consistent with a VCP hexamer complexed with six AT3 molecules (Fig. 2g). The loss of the higher molecular weight assemblies with co-transfected CΔ, indicates that it at least dynamically destabilizes what are otherwise AT3:VCP double hexamers.

We next evaluated the in vitro stability of VCP hexamers interacting with recombinant CΔ, AT3, or control GFP. Equimolar CΔ disassembled VCP hexamers on TEM (Supplementary Fig. 2e). However, pre-assembled AT3:VCP complexes resisted CΔ-mediated disassembly, even in 10-fold excess CΔ (Fig. 2h). The opposite followed CΔ addition to VCP before AT3, even in 10-fold excess AT3 (Fig. 2h). Both AT3:VCP hexamers and CΔ:VCP disassembled hexamers are highly stable. Thus, a sufficient number of AT3 proteins (probably six: one per monomer of VCP) bind to each VCP hexamer to prevent CΔ-mediated disassembly.

### VCP co-localizes on chromatin with AT3

Chromatin immunoprecipitation sequencing (ChIP-seq) was employed to test for AT3 and VCP colocalization across the genome. Traditional formaldehyde cross-linking was sufficient for AT3 ChIP-seq (using an antibody against the amino terminus of ASPSCR1), however, VCP ChIP-seq required two sequential cross-links: first protein-to-protein, with dimethyl pimelimidate, then protein-to-DNA with formaldehyde. VCP distribution on chromatin matched AT3 very tightly in FU-UR-1 cells, ASPS-1 cells, two human and three mouse ASPS tumors (Fig. 3a, Supplementary Fig. 3a–h). Many AT3:VCP target loci clustered across the genome. The most highly enriched regions of stitched AT3-ChIP-seq peaks overlapped between cell lines and tumors (Supplementary Fig. 4a, b). AT3 peaks within the most highly enriched regions demonstrated strong co-localization for VCP, peak-for-peak (Fig. 3b, Supplementary Fig. 4c, d).

VCP-ChIP followed by quantitative polymerase chain reaction (qPCR) at a series of AT3 peaks tested the dependence upon AT3 for

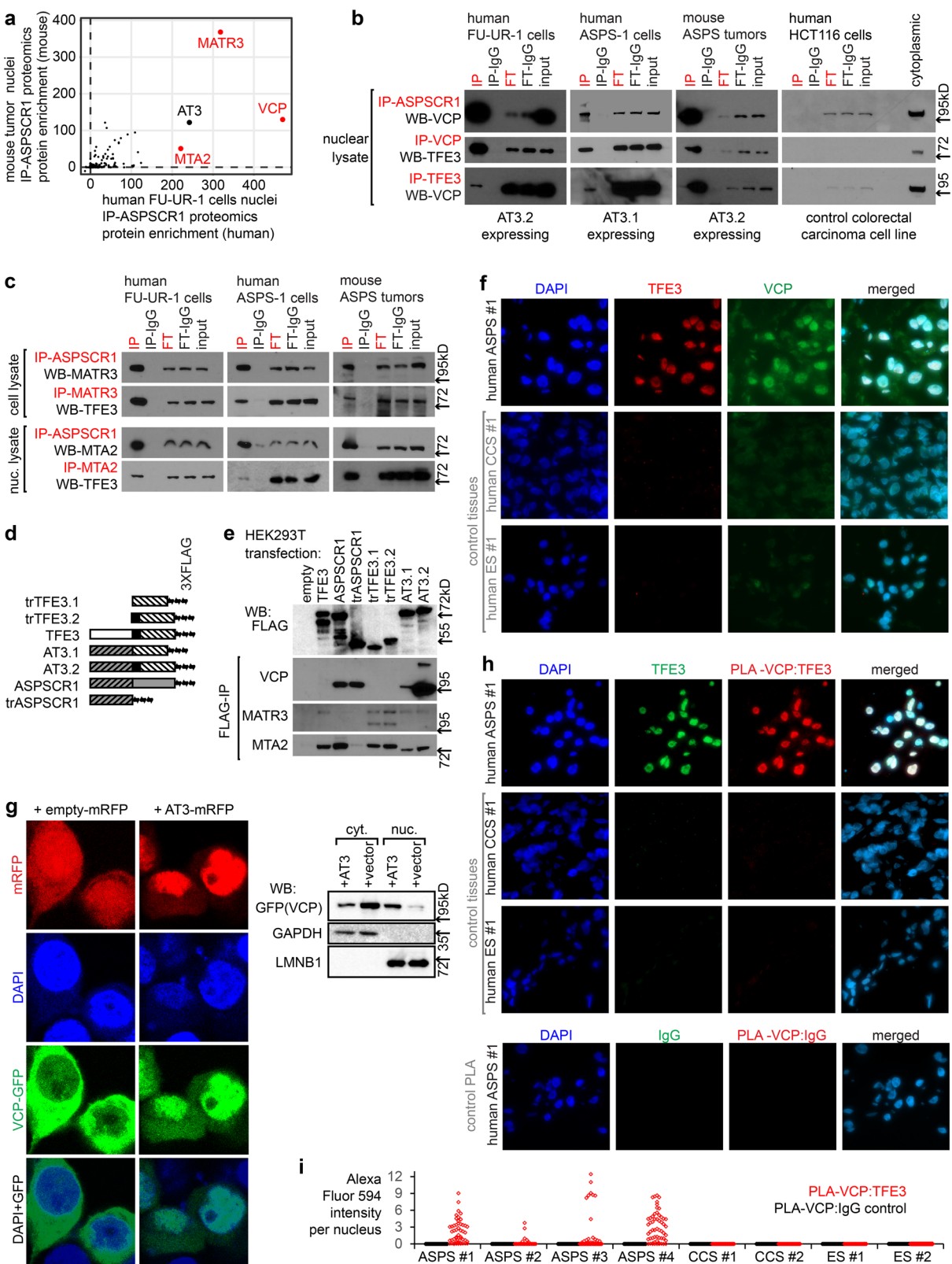

VCP presence. VCP was significantly more enriched at these loci in FU-UR-1 and ASPS-1 cells than in control HCT116 and ASKA cells (colorectal carcinoma, synovial sarcoma cell lines, respectively, lacking AT3; Fig. 3c). VCP enrichment reduced following AT3 depletion by siRNA in FU-UR-1 cells (Fig. 3d). VCP-ChIP-qPCR showed enrichment in HEK293T cells transfected with AT3.1 or AT3.2, but not controls of TFE3, ASPSCR1, or GFP (Fig. 3e).

## AT3:VCP target loci are found clustered in promoters and enhancers genome-wide

A panel of native ChIP-seq assays in five mouse ASPSs and two human ASPSs identified enhancers by monomethylated lysine 4 of histone 3 (H3K4me1), active enhancers by acetylated lysine 27 of histone 3 (H3K27ac), actively transcribed chromatin by trimethylated lysine 36 of histone 3 (H3K36me3), and transcriptionally repressed chromatin

**Fig. 1 | AT3 interacts with VCP/p97 in the nucleus of ASPS and RCC tumor cells through its ASPSCR1 portion. a** Plot of enrichment scores (defined in Methods) for homologous proteins identified on mass spectroscopy proteomics following ASPSCR1 immunoprecipitation (IP) of nuclear lysates from human FU-UR-1 cells ($n = 8 + 2$ controls IgG) and mouse ASPS tumors ($n = 7 + 3$ controls IgG) **b** IPs followed by western blots (WB) showing reciprocal AT3 (by ASPSCR1 or TFE3 antibodies) and VCP interactions in multiple tumor cell circumstances, but not control HCT116 cells (input = 10% of sample; FT = flow-through; IP-IgG and FT-IgG lanes are mock IPs with non-specific IgG; each IP-WB was repeated on a biologically independent sample; all uncropped gel images provided in Source Data file in Supplemental Information). **c** IP-WBs demonstrating reciprocal interactions in multiple tumor contexts between AT3 and MATR3 or MTA2 ($n = 2$ biological repeat not shown). **d** ASPSCR1, TFE3, and AT3 constructs oriented left-to-right:amino-to-carboxy. Gray background coded by *ASPSCR1*; slashes indicate amino acids 1-311. White background coded by *TFE3*; slashes indicate exon 4 through 3'-terminus; black coded by exon 3, the putative activation domain. **e** FLAG-IP-WB for MATR3, MTA2, and VCP in HEK cells transfected with FLAG-tagged constructs. Expression varied on $n = 3$ biological repeats: more AT3.2 was expressed in the iteration of the experiment depicted. **f** Fluorescence photomicrographs of a human ASPS tumor and control clear cell sarcoma (CCS) and Ewing sarcoma (ES) tumors stained with DAPI, TFE3 (detecting AT3), and VCP antibodies. (panel = 100 μm square; $n > 4$ fields per each of $n = 4$ tumors/controls). **g** Fluorescence images of HEK293T co-transfected with VCP-GFP and either ASPSCR1-TFE3-mRFP or mRFP alone. (panel = 30 μm square). Nuclear (nuc.) and cytoplasmic (cyt.) fraction western blots (WBs) for VCP ($n > 4$ fields per $n = 2$ biological repeated experiment). **h** Fluorescence photomicrographs of proximity ligation assay (PLA) between TFE3 and VCP antibodies in the ASPS tumor and controls, with an additional IgG-PLA control. (panel=100 μm square; $n > 4$ fields per each of $n = 4$ tumors/controls). **i** Quantitative fluorescent PLA signal per nucleus was measured in (79, 57, 143, 29, 105, 81, 83, 57, 216, 271, 215, 244, 84, 59, 195, 65 nuclei for the samples listed in order.).

by trimethylated lysine 27 of histone 3 (H3K27me3). Cross-linked RNA polymerase II (RNAPol2) ChIP-seq identified active transcription. Overall, active enhancer marks flanked and RNA polymerase peaks coincided with AT3:VCP peaks. H3K27me3 enrichment was absent from AT3:VCP enriched regions. These phenomena were observed at both promoter and distal AT3:VCP peaks (Fig. 4a–e, Supplementary Fig. 5a–f; AT3 overlap with H3K27ac vs. random: $\chi^2 =$ human 17144, mouse 11062; AT3 overlap with H3K4me1 vs. random: $\chi^2 =$ human 7198, mouse 19651; AT3 overlap with RNAPOL2 $\chi^2 =$ human 4368.9, mouse 11268; for all, df = 1 and $p$-value $< 2.2 \times 10^{-16}$).

AT3:VCP peaks in highly enriched regions annotate nearest genes that are generally transcribed at a higher level in tumors than in muscle from both mice and humans (Fig. 4a, Supplementary Fig. 5c). Genes nearest to three or more peaks in these regions had significantly higher differential expression in tumors over muscle than genes nearest to only 1 or 2 peaks (Fig. 4f). However, not all highly targeted genes were apparently overexpressed. To ask if AT3:VCP exclusively activates transcription, AT3 and/or VCP were depleted by small interfering RNA (siRNA) in FU-UR-1 and ASPS-1 cells (Supplementary Fig. 5g, h). Both up and downregulated genes by AT3 depletion were direct targets by proximity to highly enriched ChIP-seq regions (Fig. 4g, Supplementary Fig. 5i). These direct target genes with expression reduced by AT3 depletion were largely shared between the two cell lines, and enriched for mitosis and cell cycle pathways in Reactome Enrichr analysis (Fig. 4h, Supplementary Table 1).

## VCP presence, hexamer-assembly, and enzymatic function impact AT3-related transcription

VCP depletion rendered highly correlated differential expression with AT3 depletion among direct target genes (Fig. 5a, b, Supplementary Fig. 6a–c). These reverse genetic approaches suggest that VCP presence at AT3:VCP targets enables the noted gene regulation. The inverse approach of exogenous expression was used to interrogate the transcriptional consequences of altering AT3:VCP dynamics in HEK293T cells. HEK293T cells are more easily transfected than FU-UR-1 or ASPS-1 cells and—as embryonic kidney cells—represent reasonable cells of origin to model RCC that expresses AT3. HEK293T cells expressing AT3 have transcriptional profiles similar to FU-UR-1 cells[7]. Reverse transcriptase-generated complementary DNA followed by qPCR (RT-qPCR) was used to test expression changes across a panel of control and target genes (defined by AT3:VCP-ChIP-seq and reduced expression upon AT3 or VCP depletion in FU-UR-1 and ASPS-1 cells).

AT3.1, AT3.2, or TFE3 in full-length or truncated forms induced expression of the panel of target genes, but not control genes in HEK293T cells. Co-transfected VCP overexpression enhanced target gene expression only when combined with AT3.1 or AT3.2, but neither alone nor combined with full-length or truncated TFE3 (Fig. 5c, Supplementary Fig. 6d). Although native VCP expression is high in cytoplasm, its exogenous overexpression enhanced AT3-driven transcription, arguing that they collaborate.

Next, CΔ overexpression to disassemble VCP hexamers was employed to test the importance of hexamer structure in AT3:VCP-mediated transcription. HEK293T cells transfected with GFP control, AT3.1, AT3.2, TFE3, or each co-transfected with CΔ were subjected to RNA-seq. Genes at least 8-fold (and significantly) upregulated over GFP control by single addition of AT3.1, AT3.2, or TFE3 were selected (Wald statistical test, Benjamini-Hochberg adjustment, $p$-adjusted <0.01). Co-transfection of CΔ reduced expression of these genes when combined with AT3.1 or AT3.2 (Fig. 5d). Combined with control TFE3, CΔ impacted transcription insignificantly and balanced in each direction.

CΔ-mediated hexamer disassembly unavoidably abrogates VCP enzymatic function (Supplementary Fig. 6e)[19]. CB-5083, a VCP-specific, potent, small molecule ATPase inhibitor[20] blocks enzymatic function without disassembling hexamers. Cell Titer Glo assays identified inhibitory concentrations of CB-5083 in both FU-UR-1 and ASPS-1 cells as well as control cancer cell lines HCT116 (colorectal carcinoma), ASKA (synovial sarcoma), and two cell lines that natively express other TFE3 fusions, UOK146 that expresses PRCC-TFE3 and UOK109 that expresses NONO-TFE3 (Supplementary Fig. 6f). The transcriptional impact of CB-5083 (applied to FU-UR-1 and ASPS-1 cells for 48 h) correlated with siRNA-mediated depletion of AT3 or VCP, but less strongly than each depletion with one another (Fig. 5e, f, Supplementary Fig. 6g). This suggested that a portion of VCP's transcriptional impact is independent of its physical interaction with AT3 on chromatin, possibly reflecting the transcriptional activation of metabolic stress-response genes (many of which are TFE3 targets), induced by cytoplasmic VCP inhibition in most cell types[21].

CB-5083 was next applied to the HEK293T model, providing a TFE3-overexpression control for the AT3-induced transcriptional state. Many of the genes that were 8-fold upregulated by each of the three transcription factors (AT3.1, AT3.2, TFE3) were further upregulated by CB-5083 in TFE3-transfected cells (Fig. 5g). Because VCP does not interact with native TFE3 (Fig. 1e) and VCP is minimally present in the HEK293T nucleus without AT3 (Fig. 1g), this increased upregulation highlights the role of cytoplasmic VCP inhibition on transcription. In contrast, cells transfected with AT3.1 or AT3.2 consistently showed transcriptional downregulation of these same genes by CB-5083 (Fig. 5h). That CB-5083 impacts AT3 transcription oppositely from TFE3 transcription suggests two ideas. First, VCP inhibition in the presence of AT3 alters specific AT3:VCP targets for transcription. Second, at least at these genes, AT3:VCP transcriptional consequences override those of cytoplasmic VCP inhibition (which certainly still occur in AT3-expressing cells upon exposure to CB-5083). This

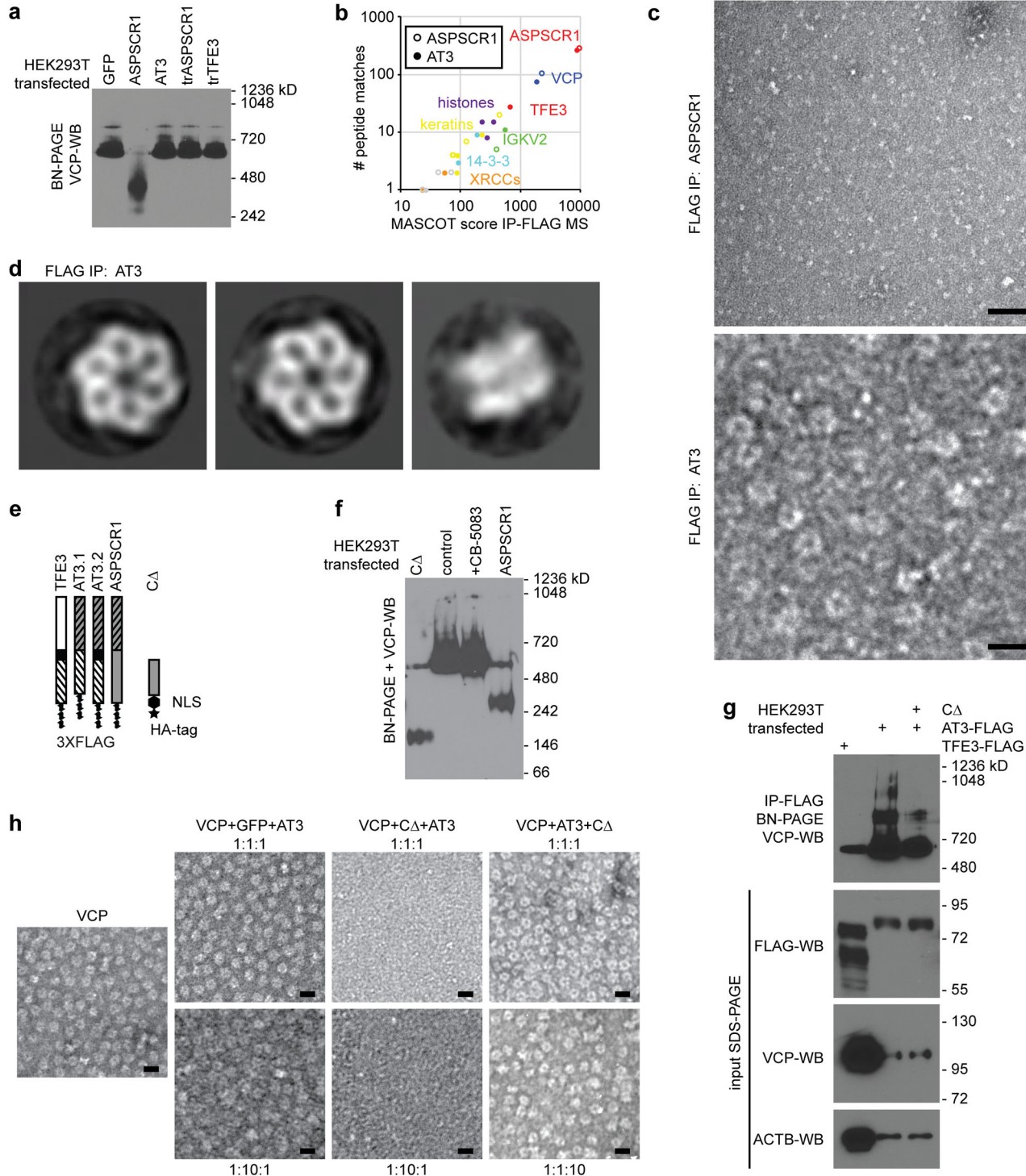

**Fig. 2 | AT3 interacts with hexameric assemblies of VCP. a** WB for VCP after BN-PAGE of whole cell lysates from HEK293T cells after transfection with control GFP, ASPSCR1, AT3, truncated ASPSCR1 (trASPSCR1, the portion included in AT3) or truncated TFE3 (trTFE3, the portion included in AT3). VCP hexamer size is ~582 kD (*n* = 3 biological repeats performed with similar result). **b** Mass spectrometry results after FLAG-IP from HEK293T cells transfected with tagged ASPSCR1 or AT3, showing VCP predominant in both, other than the peptides aligned with the FLAG-tagged bait itself, noted in red as ASPSCR1 or both ASPSCR1 and TFE3. **c** Negative stain TEM micrographs of FLAG-IP eluates after ASPSCR1 or AT3 overexpression in HEK293T cells (bars = 20 nm; biological repeats *n* = 3). **d** Reference-free 2D class averages of negatively stained particles recovered from FLAG-AT3 co-IP reveal intact VCP hexameric assemblies. Left and middle, VCP top or pore views (3620 and 2774 particles in each class, respectively); right, VCP side view (182 particles; box length and width = 25 nm). **e** Schematic of constructs of each type of AT3 as well as CΔ, a truncated portion of the carboxy half of ASPSCR1, not included in AT3 (HA human influenza hemagglutinin tag; NLS nuclear localization signal). Orientation from top-to-bottom is amino-to-carboxy termini. **f** VCP WB after BN-PAGE following transfection of HEK293T cells with ASPSCR1, CΔ, or an empty vector control as well as the last of these transfections treated subsequently with VCP inhibitor CB-5083 (biological repeats *n* = 3). **g** VCP WB after BN-PAGE following IP for TFE3-FLAG or AT3-FLAG with or without co-transfected CΔ, along with denaturing gel blots of input proteins below (biological repeats *n* = 3). **h** Negative stain TEM micrographs of in vitro mixing of VCP with control green fluorescent protein (GFP) or CΔ, with AT3, added in the order of the molar ratios listed (bars = 20 nm; biological repeats *n* = 2).

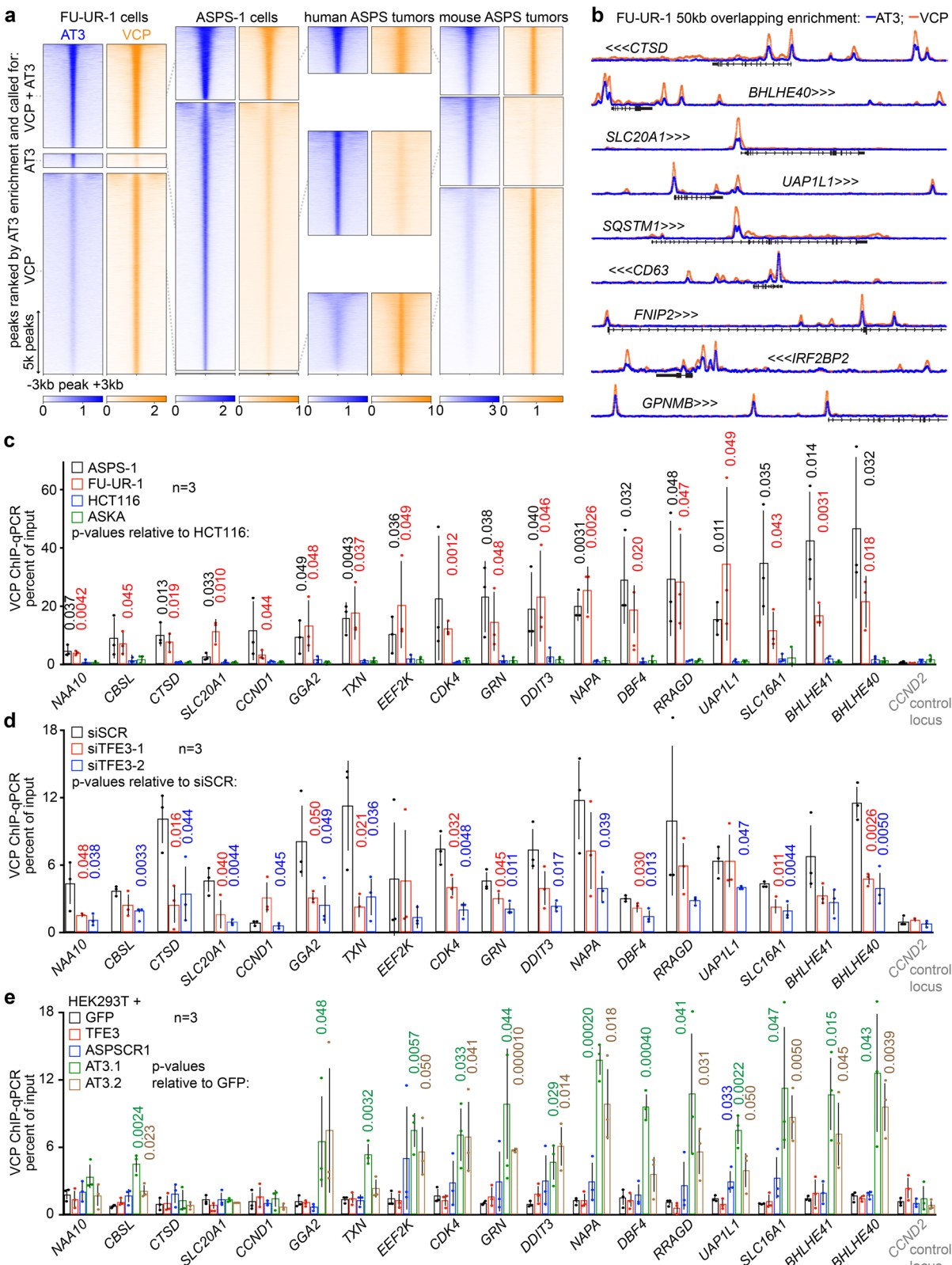

phenomenon was confirmed by RT-qPCR for AT3:VCP targets in the panel of cell lines, after confirming that the alternative TFE3 fusions did not co-IP with VCP (Supplementary Fig. 6h, i). These results prompted a working model for the effects of VCP loss, hexameric disassembly, or enzymatic inhibition on AT3 target gene transcription (Fig. 5i)

For an orthogonal approach to depleting VCP enzymatic activity with an inhibitor, we next tested the impact of adding an enzymatically

inactive VCP to AT3-mediated transcription in the HEK293T model. Transfected with AT3.1 or AT3.2, VCP$^{E305Q}$ (D1 Walker B mutation) enhanced transcription of some target genes (Fig. 5j), similar to wild-type VCP. At other loci, only wild-type VCP enhanced AT3-mediated transcriptional upregulation significantly. VCP may therefore have both structural/scaffolding and enzymatic functions that impact transcription differently at different AT3:VCP target sites.

**Fig. 3 | VCP co-localizes on chromatin to target loci of AT3. a** Heatmaps of overlapping enrichment in ChIP-seq with antibodies against AT3 and VCP in human FU-UR-1 cells, ASPS-1 cells, two human ASPS tumors, and 3 mouse ASPS tumors. Each heatmap is centered on peaks across the genome, sorted by AT3 enrichment, representing at least 2 replicates. **b** Example tracks with overlaid ChIP-seq enrichments of AT3 and VCP at selected highly enriched target genes (See also Supplementary Fig. 4a). Tracks scaled together 0 to 15 reads per million (RPM). **c** Graph depicting mean ± standard deviation as well as raw value points of fold-enrichment of double cross-linked VCP ChIP over input chromatin by quantitative polymerase chain reaction (qPCR) for two AT3-expressing cell lines (ASPS-1 and FU-UR-1) and two control cell lines (HCT116, colon cancer; ASKA, synovial sarcoma) at a panel of selected target genes and one negative control locus (*CCND2*). Two-tailed Student's *t*-test generated *p*-values as indicated. **d** Similar graph of VCP ChIP enrichment over input for FU-UR-1 cells exposed for 48 h to control *siSCR* or one of two *siTFE3*s directed against AT3. **e** Similar graph of VCP ChIP enrichment over input for HEK293T cells transfected with GFP control, TFE3 control, ASPSCR1 control, AT3.1 or AT3.2.

## AT3:VCP interactions impact target gene transcription by enhancer modulation

H3K27ac-HiChIP, a method of high-resolution chromatin conformation capture[22] exploited the observation that AT3:VCP ChIP-seq peaks consistently co-localized with H3K27ac (Supplementary Fig. 7a–d). To examine promoter versus enhancer chromatin associations, each AT3:VCP enriched peak in ASPS-1 and FU-UR-1 was characterized by its location within a promoter and/or its association by H3K27ac-HiChIP loops to other promoters. Each target gene was assigned a class (promoter bound only, or looped enhancer to promoter bound) and an aggregate score of the AT3 and VCP enrichments within the 1 kb surrounding each of its associated VCP peak centers. AT3 and VCP enrichments at looped enhancer peaks were multiplied by a normalized loop score before being added to the aggregate for a given target gene. These correlated strongly between AT3 and VCP for each cell line, especially for enhancer-targeted genes, and between the cell lines (Fig. 6a, b, Supplementary Fig. 7e, f).

AT3 depletion correlated better with VCP depletion for transcriptional changes among enhancer-targeted genes than among promoter-targeted genes (Fig. 6c). However, enhancer-targeted genes showed slightly poorer correlation between the impact of VCP depletion and enzymatic inhibition with CB-5083 (Fig. 6c). This suggests a scaffolding impact of the presence of VCP on looped enhancers that is partly distinct from the co-localization of its enzymatic activity at AT3:VCP-bound promoters.

Principal component analysis was performed following depletion of AT3, VCP or both (by siRNA) among the top AT3:VCP promoter- and enhancer-targeted genes. VCP depletion had specific impacts on some genes, comprising principal component 2, likely due to disruption of its cytoplasmic function (Fig. 6d). Genes with coefficients to principal component 1 greater than 0.05 (Supplementary Table 2) showed that AT3 knock-down was generally more impactful than VCP, but in a similar direction (Fig. 6e). Combination dual knock-down demonstrated no additive or synergistic reduction in their expression beyond AT3 depletion alone.

We next tested if AT3:VCP interactions drive—rather than merely associate with—these chromatin conformations. VCP depletion by siRNA for 6 days dramatically reduced the AT3:VCP-peak-associated loops (Fig. 6f–i, Supplementary Fig. 7f–h). In contrast, control HiChIP loop enrichments, from H3K27ac ChIP-seq peaks not associated with AT3:VCP, were significantly increased or remained unchanged (Fig. 6g–i).

48-hour depletion (by siRNA) of VCP or AT3 in FU-UR-1 cells was followed by H3K27ac ChIP-seq to test for possible H3K27ac depletion as a driver of the H3K27ac-HiChIP loops loss. This quantitatively tested the dependence of the enhancer character of these target loci on the presence of AT3:VCP. Following each knockdown, H3K27ac ChIP-seq enrichment was significantly reduced at AT3:VCP-associated, looped loci, in contrast to non-AT3:VCP-associated looped loci (Fig. 6j, Supplementary Fig. 7i, k).

## Physiological function of AT3-expressing cancer cells depends on VCP in vitro

Reverse genetic approaches examined the impact of AT3:VCP on proliferation of FU-UR-1 cells exposed to CFSE dye, which depletes each cell division (Supplementary Fig. 8a), detected by flow cytometry. FU-UR-1 proliferation was blunted by depletion of AT3, VCP, or both (Fig. 7a). Combination siRNA applications achieved no additive reductions in proliferation, suggesting that AT3 and VCP engage in the same pathway related to proliferation.

AT3 or VCP depletion by siRNA for 48 h prior to low-density plating blocked colony formation in FU-UR-1 and ASPS-1 cells (Fig. 7b). Cells exposed to controls (*siSCR* or an *siTFE3* that targeted a portion of *TFE3* excluded from *AT3*) formed colonies. CB-5083 similarly blunted colony formation in low-density plating of FU-UR-1 and ASPS-1 cells (Supplementary Fig. 8b).

In CFSE dye depletion, the more penetrant CB-5083 reduced proliferation further, when added to knock-down of AT3 or VCP (Fig. 7c, d). However, knock-down of AT3 or VCP added to CB-5083 also further blunted proliferation over CB-5083 alone. The effects of VCP protein presence on proliferation are therefore not entirely due to its enzymatic activity.

In counterpart to knock-down experiments, we next tested the impact of AT3 expression on proliferation, with or without over-expression of VCP in *Rosa26-LSL-AT3* mouse embryonic fibroblasts. These conditionally express a physiologic (and tumor-inducing) level of AT3.2 upon exposure to TATCre (Fig. 7e). Transcriptional activation of *AT3* or overexpression of VCP by transfection increased proliferation over control GFP alone, but their combination further amplified proliferation (Fig. 7f).

## Physiological function of AT3-expressing cancer cells depends on VCP in vivo

Lentivirally delivered IPTG-inducible shRNAs for the same sequences achieved longer-term AT3 or VCP depletion (Supplementary Fig. 8c). IPTG-induced AT3 or VCP depletion blocked tumor growth in xenografted FU-UR-1 cells in NRG mice (Fig. 8a–c). Control FU-UR-1 xenografts rapidly developed tumors that histologically resembled RCC and ASPS (Supplementary Fig. 8d).

CB-5083 treatment of FU-UR-1 xenografted mice by oral gavage cytoreduced tumors (Fig. 8d). Gut toxicity of the poorly soluble gavage slurry has challenged pre-clinical CB-5083 development. Although unrelated to VCP inhibition directly, this toxicity impacts drug tolerance, requiring reduced dosages and treatment frequencies. Mice initially treated at 100 mg/kg for 4 days per week tolerated no more than 8 days on treatment, but showed significantly reduced xenograft growth. Longer-term reduced-dosage therapy also blunted xenograft growth. Ki-67 immunofluorescence showed reduced proliferation in CB-5083-treated xenografts compared to vehicle-treated tumors (Fig. 8e).

CB-5083 experiments were repeated in an ASPS patient-derived xenograft, also at a reduced dose. This model also demonstrated significant blunting of tumor growth by weekly luciferase measurements (Fig. 8f, Supplementary Fig. 8e).

CB-5083 or vehicle was also administered in *Rosa26-LSL-AT3/CreER* mice, developing perivascular tumors in the brain[5]. The brain is an uncommon site for primary ASPS, but a frequent and dangerous site for ASPS metastasis that presents particular challenges to systemic treatments, affording this pre-clinical model specific relevance. Assessed by weekly brain MRI, as a tumor reached a volume of 10 mm³, its host mouse was randomized to CB-5083 or vehicle treatments, 4

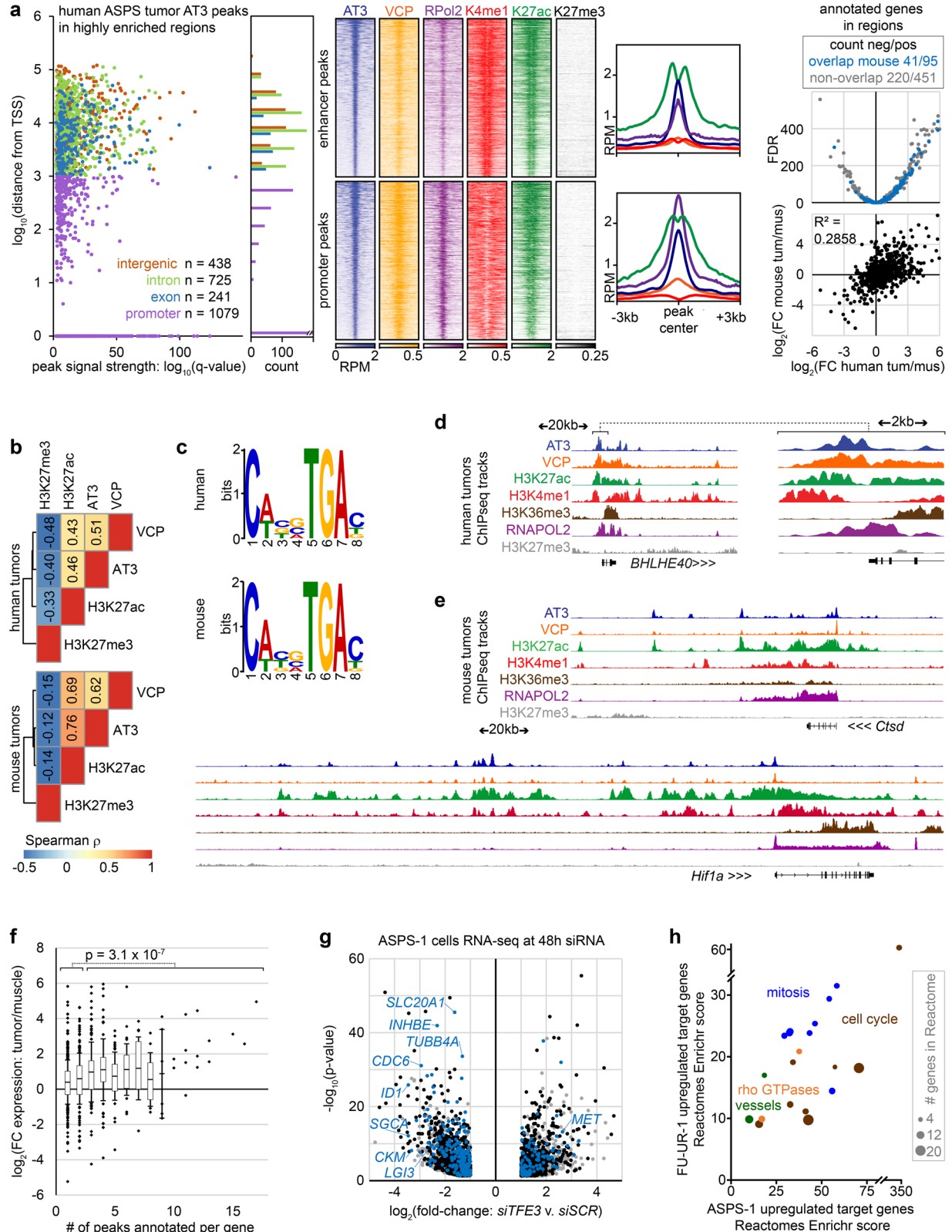

times weekly. Mice receiving CB-5083 survived longer with more slowly growing tumors (Fig. 8g, Supplementary Fig. 8f, g). RT-qPCR showed that AT3:VCP targets were more highly expressed in mouse AT3-initiated tumors treated with vehicle than in control mouse tumors (osteosarcoma, synovial sarcoma); CB-5083 for 4 days immediately prior to harvest also reduced expression of most of these (Fig. 8h, Supplementary Fig. 8h).

## Discussion

The strength and specificity of the AT3:VCP interaction across chromatin genome-wide elevate hexameric VCP to the level of a true and perhaps necessary co-factor for AT3. Negative stain 2D class averages identified VCP hexamers stably interacting with AT3, unperturbed by in vitro addition of the disassembly factor CΔ. Loss of VCP or even loss of its hexameric assembly in cells reduced the transcriptional impact

**Fig. 4 | AT3:VCP interaction localizes to the promoters and enhancers of target genes. a** Graph and annotation histogram for AT3 ChIP-seq peaks called in 2 human ASPS tumors within highly enriched regions (definition Supplementary Fig. 4a). RPM heatmaps for distal (putative enhancer; 1404 peaks) and proximal promoter peaks (1079 peaks; H3K4me1 window ±10 kb, others ±3 kb), and enrichment plots (H3K27me3 non-significant), then mean differential expression of nearest genes in human tumors compared to muscle samples by RNAseq (dataset GSE54729, FDR = false discovery rate $q$-value), noting genes also annotated in mouse highly enriched AT3 regions and correlation human to mouse differential expression tumors over muscle. **b** Spearman rank correlation coefficients between AT3 ChIP-seq enrichment genome-wide compared to VCP, H3K27ac and H3K27me3, (For human tumors, S = $2.81 \times 10^{13}$, $3.24 \times 10^{13}$, $8.45 \times 10^{13}$, respectively; all $p$-values < $2.2 \times 10^{-16}$). **c** Dominant motifs in peaks co-enriched for AT3, H3K27ac, and RNAPOL2. **d** Example ChIP-seq tracks from human tumors ($n = 2$, pooled) showing the region and 10-fold-magnified view (genomic distance scale noted for each). Vertical track scales are AT3: 0–4, VCP: 0–2, H3K27ac: 0–5, H3K4me1: 0–1, H3K36me3: 0–1, RNAPOL2: 0–6, H3K27me3: 0–0.2. **e** Example ChIP-seq tracks for *Ctsd* and *Hif1a* in mouse AT3-induced tumors. Vertical track scales AT3: 0–10, VCP: 0–4, H3K27ac: 0–9, H3K4me1: 0–2, H3K36me3: 0–2, RNAPOL2: 0–10, H3K27me3: 0–2. **f** Plots of mean fold-change expression in human ASPS tumors over muscle (25th to 75th percentile boxes and error bars of standard deviation, outliers depicted individually; $n = 5$, 3, respectively) for each number of peaks associated by nearest-gene annotations. (two-tailed heteroscedastic $t$-test $p$-value comparing 1–2 peaks to 3 or more). **g** Differential gene expression following 48-h siRNA depletion of AT3 in ASPS-1 cells. Blue dots represent genes annotated by highly enriched regions (from Supplementary Fig. 4a). Genes noted in black are those annotated by AT3 peaks generally. Gray dots have no associated AT3 ChIP-seq peaks. (Wald test $p$-values, Benjamini-Hochberg correction.) **h** Enrichr plot of Reactome gene sets enriched in both FU-UR-1 and ASPS-1 direct target genes of highly enriched areas and downregulated on AT3 depletion in each cell line. Specific Reactomes were assigned to general categories.

of AT3. VCP depletion abolished the AT3-associated three-dimensional organization of chromatin. Depletion of either efficiently blunted proliferation and tumorigenesis. Notably, a VCP-TFE3 fusion was identified in another human mesenchymal cancer[23]. That fusion included only the D2 ring from VCP, which can assemble into hexamers, but lacks segregase function. A principal question for the next experiments in AT3:VCP-regulated transcription will include how much, and by what mechanisms do the enzymatic segregase and scaffolding functions of hexameric VCP impact transcription?

AT3 is considered to be a transcription-activating factor at target genes, but some directly targeted genes were transcriptionally repressed. For example, the *MET* oncogene is heavily targeted by AT3:VCP, but increases expression upon knock-down of either in FU-UR-1 and ASPS-1 cells (Figs. 4g, 5a, b, Supplementary Fig. 5i). Prior papers concluding that AT3 upregulates *MET* expression did not perform AT3 depletion in natively AT3-expressing cells[24–26]. By what mechanism AT3:VCP represses expression of some targets deserves additional attention.

Although many AT3:VCP targets are shared across cellular contexts and even species, some differ, perhaps highlighting subtle differences between RCC and ASPS. For example, there are highly enriched AT3:VCP ChIP-seq peaks around the *Hif1a* locus in mouse ASPS tumors (Fig. 4e) and around the homologous *HIF1A* locus in ASPS-1 cells and human ASPS tumors. No AT3:VCP peaks were identified in FU-UR-1 cells near *HIF1A*, as previously reported[7].

In the transcription assays in HEK293T cells (Fig. 5d, h), the impact of CB-5083 and CΔ were noticeably different between AT3.1 and AT3.2, the effects being more pronounced with AT3.1. VCP ChIP-qPCR found more VCP at target loci in ASPS-1 cells (AT3.1 expressing) than in FU-UR-1 cells (AT3.2 expressing, Fig. 3c, e). With the retained TFE3 activation domain, AT3.2 may be less strictly dependent on VCP for transcriptional impact.

The identification of VCP as co-factor to AT3 strengthens the connection between ALS and fusion-associated sarcomas. *FUS* and *EWSR1*, participants in other sarcoma fusion oncogenes, are both implicated in familial ALS[27–29]. Mutations in *VCP*, *MATR3*, *hnRNPU*, *hnRNPA1*, *hnRNPA2B1*, *FUS*, *TARDBP* and genes coding for other AT3-interactors (Supplementary Fig. 1a) also drive familial ALS[30–33]. Transcriptional impact of EWSR1 fusions depends on multi-valent interactions with its prion-like N-terminal domain[29,34]. The multi-valency of AT3 interacting with VCP hexamers is similarly critical to its oncogenic transcriptional impact (Fig. 5d).

How does VCP enzymatic segregase activity contribute to AT3-mediated transcription? Because VCP ATPases depend on hexamerization, scaffolding and enzymatic functions are difficult to isolate experimentally. Inhibition of enzymatic activity alone mirrored the impact of VCP depletion at least partly (Figs. 5e, f), providing an excellent tool to study this further. VCP segregase substrates at AT3:VCP target loci need to be identified. A few nuclear VCP substrates have been identified previously[35–37]. Yeast VCP (Cdc48) transcriptionally activates some target genes by segregating ubiquitinated histones[38,39]. Mammalian cell transcriptional transitions between cell cycle phases can similarly depend on VCP[40]. VCP targeted to AT3-binding-defined chromatin regions may remove repressive factors. Notably, VCP enzymatic inhibition more closely recapitulates VCP depletion at promoter-targeted than at enhancer-targeted genes (Fig. 6c).

The prospect of advancing VCP inhibitors as therapy for ASPS or RCC remains challenging. Phase I CB-5083 clinical trials were terminated due to off-target toxicities[41]. CB-5083 reduced growth rates of AT3-initiated mouse tumors and cell line and patient-derived xenografts, but mice spent half of each week receiving no drug, due to off-target toxicity concerns. There are other VCP inhibitors in development and targeted protein degradation may provide additional opportunities[42–44]. Most importantly, as the identification of menin as a co-factor for MLL fusions did for leukemias, VCP identified as a co-factor for the AT3 fusion oncoprotein will immediately widen the avenues of investigation into both biology and therapy for ASPS and Xp11-rearranged RCC.

## Methods
### Mouse models
All mouse experiments were performed under the auspices of the Institutional Animal Care and Use Committees at the University of Utah or the University of Calgary and in accordance with international legal and ethical guidelines. The mouse genetic model of alveolar soft part sarcoma was previously described[5]. Tumors were harvested at necropsy and snap frozen in liquid nitrogen. For the FU-UR-1 xenograft experiments, the subcutaneous flanks of NRG mice were injected with 1 million cells in matrigel. Caliper tumor measurements were performed at least thrice weekly. The patient-derived xenograft was from a resected breast metastasis of an ASPS in an adolescent female (described below in detail). Cells transduced with firefly luciferase were injected into the gastrocnemius muscles of 6–8-week-old NOD/SCID mice, checked for tumor development at 60 days by Xenogen IVIS imaging, then randomized to treatment with CB-5083 or vehicle. All mice were housed in standard 14-h/10-h light-dark cycle rooms at 18–23 degrees Celsius temperature and 40–60 percent humidity.

### Human tumor samples
Frozen samples were utilized in de-identified fashion from repositories collected with patient consent and institutional review board approval at the University of Utah, University of British Columbia, and the University of California at San Francisco. The tissue microarray was provided from MD Anderson Cancer Center. Samples of both male and female sex were included, but blinded as to which was which.

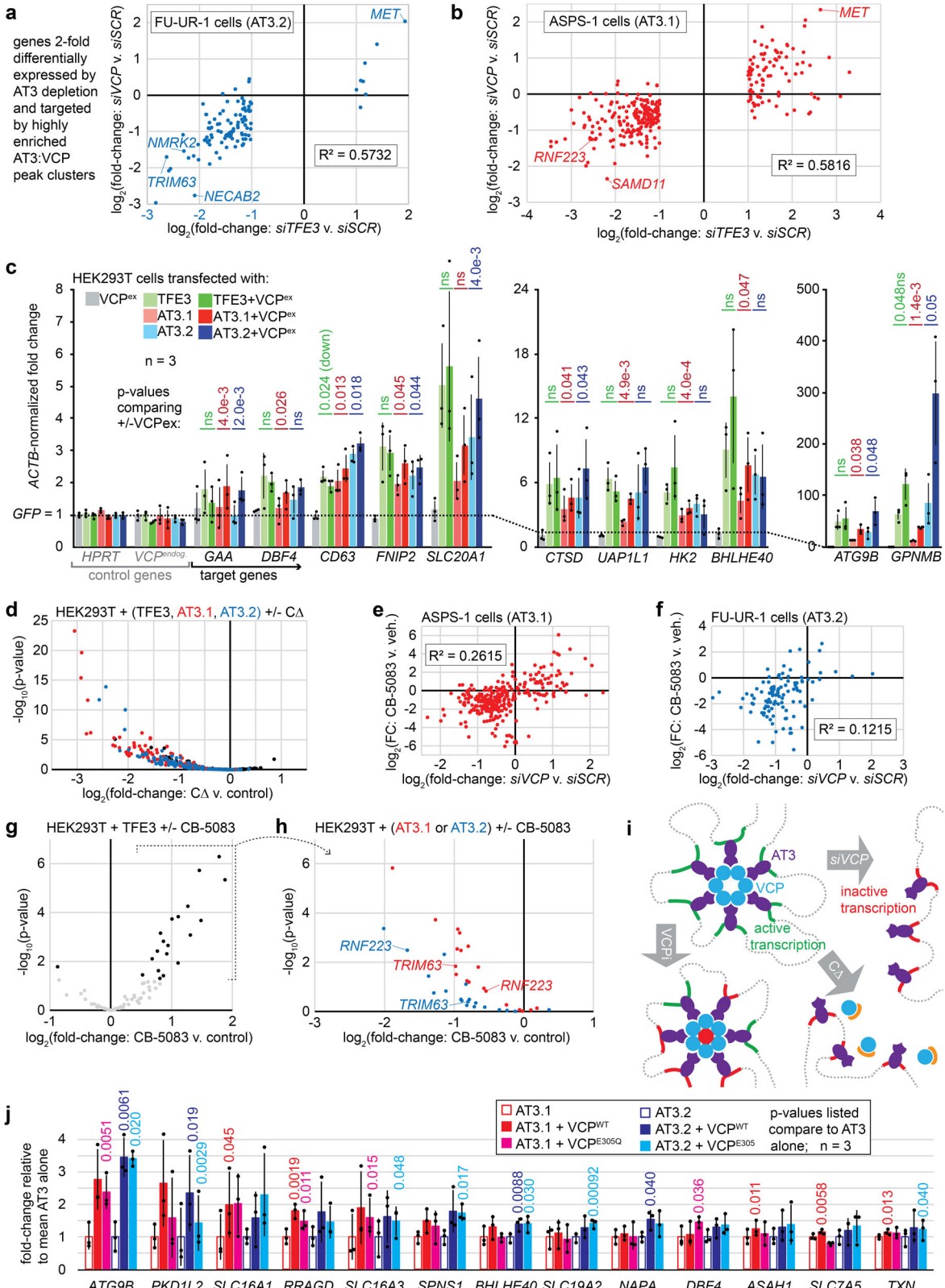

## Human cell lines

HEK293T cells were obtained from ATCC and were maintained in DMEM supplemented with 10% FBS.

FU-UR-1 cells were obtained from Marc Ladanyi at Memorial Sloan-Kettering and were maintained in Dulbecco's Modified Eagle Medium (DMEM) plus Ham's F12 media (1:1 volumetric ratio) supplemented with 10% FBS.

ASPS-1 cells were obtained from the Division of Cancer Treatment and Diagnosis, NCI, NIH, USA and were maintained in DMEM/F12 (1:1 ratio) media supplemented with 10% FBS.

Each cell line was tested at least annually by STR sequencing to confirm stability and identity of the cells in prolonged culture. FU-UR-1 is from a patient host of male sex and ASPS-1 from a patient host of female sex.

**Fig. 5 | VCP presence, hexamerization, and ATPase activity impact AT3-related transcription. a** Correlation of fold-change (FC; to control) FU-UR-1 RNA-seq transcription comparing siRNAs that deplete VCP or AT3 among target genes (highly enriched AT3 region genes with >2FC by AT3 depletion. ($n = 3 \times 2$ siRNAs each for VCP and AT3, pooled.) **b** Similar targets correlation for ASPS-1 (also $n = 3 \times 2$ pooling). **c** RT-qPCR for target genes (controls *HPRT*, *VCP*) for HEK293T expressing TFE3, type 1 or 2 ASPSCR1::TFE3 (AT3.1, AT3.2), each with or without exogenous VCP (VCP^ex) overexpression. ($n = 3$, pooled; mean ± standard deviation; homoscedastic two-tailed *t*-tests comparing added VCP^ex to each alone.) **d** Genes plotted had at least 8-fold upregulated expression by RNA-seq for TFE3, AT3.1, and AT3.2 relative to control *GFP* in the absence of CΔ in HEK293T cells. The plot indicates log transformed *p*-values and FC comparing each transcription factor with co-transfected CΔ to that without CΔ. Co-transfected CΔ to HEK293T-cells changes TFE3 targets in both directions, mostly insignificantly. CΔ significantly downregulates most AT3.1/AT3.2 targets ($n = 3$ each group, homoscedastic two-tailed *t*-tests). **e** Less pronounced correlation between the differential expression by RNA-seq in ASPS-1 from CB-5083 relative to vehicle at 48 h ($n = 3$ for each) to siRNAs depleting VCP relative to control at 48 h ($n = 3 \times 2$ pooled for each). **f** The same for FU-UR-1 cells (for CB-5083 experiment, $n = 2$ for each). **g** Plot of log-transformed *p*-values and FCs of CB-5083 or DMSO treated HEK293T cells transfected with TFE3. The genes included were 8-fold upregulated by TFE3, AT3.1 and AT3.2 over GFP control transfection. Most significant changes showed further upregulation by CB-5083. ($n = 3$, pooled, homoscedastic two-tailed *t*-tests). **h** Genes upregulated by CB-5083 added to TFE3 are mostly downregulated by CB-5083 added to AT3.1 or AT3.2. ($n = 3$, pooled, homoscedastic two-tailed *t*-tests). **i** Schematic working model where VCP hexamers interact with chromatin via AT3 to activate transcription, abrogated by VCP depletion, hexamer disassembly, or enzymatic inhibition, the last with mixed impacts. **j** RT-qPCR for target genes after TFE3, AT3.1, or AT3.2 expression in HEK293T with co-transfection of control, VCP, or VCP^E305Q (hexamer-assembling, enzymatically inactive mutant), which increases some targets' expression ($n = 3$, pooled; mean ± st.dev.; homoscedastic two-tailed *t*-tests).

## Bacterial strains

DH5α bacteria were purchased from Thermo Fisher Scientific, USA and grown in Luria Bertani broth for cloning and plasmid preparation.

For overexpression of plasmids for purification of proteins, BL21(DE3)RIL bacteria were purchased from Agilent Technologies (Santa Clara, CA). Expression cultures were grown in ZY autoinduction media for purification of overexpressed proteins[45].

## Statistical analysis

Statistical comparisons between two groups were performed using two-tailed Student's *t*-test in Graphpad Prism software 7.0, and statistical significance was set at $p < 0.05$ or $0.01$ as indicated in the figure legends. Data are presented as mean ± standard deviation. The sample size was determined by the variance in results of preliminary experiments. The number (n), as indicated in the figure legends, represents the number of biological replicates, not multiple measurements of the same sample. Other statistical methods used are clearly depicted in each figure legend. Genomics-related statistics were performed as described in the pertinent analysis sections, in the Methods section.

## Co-Immunoprecipitation (Co-IP)

Nuclear complex Co-IP kit (Active Motif) was used for the Co-Immunoprecipitation according to the manufacturer's instructions for most protein interactions. The successful separation of nuclear contents was checked directly on the IP samples as having only a single band at the AT3 size (not ASPSCR1 size) and not cross reactive with carboxy-terminus ASPSCR1 antibody. Co-IP for MATR3, specifically, was performed instead as previously described[46]. Briefly, one 10 cm plate of 90% confluent cells was washed with 5 ml of cold PBS and scraped in 2 ml cold PBS on ice. Scraping was done twice from the each plate and collected in a 15 ml conical tube. Cells were collected by centrifugation at $500 \times g$ for 5 min at 4 °C. Cells were lysed for 30 min in a buffer having 0.5% NP40, 150 mM NaCl, 50 mM Tris pH7.5, 1 mM EDTA supplemented with protease inhibitor cocktail on ice. Supernatant was collected after centrifugation at 13,000 rpm for 15 min at 4 °C. An equal amount of lysate was incubated with 4 μg of anti-ASPSCR1/Rabbit IgG overnight at 4 °C. Antibody-protein complexes were pulled down using 50 μL of protein-G Dynabeads (Invitrogen). Beads were washed 3 times with 300 μL of PBS supplemented with 0.02% Tween-20.

This protocol was also used for the immunoprecipitation from the AT3.2 / AT3.1 / trTFE3.2 / trTFE3.1 / trASPSCR1 / ASPSCR1-FL / TFE3-FL / IRESAcGFP transfected HEK293T cells.

For co-immunoprecipitated complexes imaged by negative stain TEM, HEK cell pellets with expressed corresponding C-terminally FLAG-tagged ASPSCR1 variants were resuspended with five volumes of IP buffer (50 mM HEPES-KOH pH 6.8, 100 mM KOAc, 10 mM Mg(OAc)$_2$, 5% glycerol, 0.1% Igepal CA-630, 1 mM DTT, 3 units/mL benzonase, and a protease inhibitor cocktail containing 0.5 μg/mL aprotinin, 0.7 μg/mL pepstatin, 0.5 μg/mL leupeptin, and 17 μg/mL PMSF). The suspensions were then transferred to a Dounce homogenizer and cells were lysed with 10–20 strokes[47,48]. The lysates were then clarified by centrifugation at $17,000 \times g$ for 20 min. The supernatants were incubated with equilibrated anti-FLAG M2 affinity gel (Sigma) at 4 °C for an hour. Unbound material was separated from affinity gel by centrifugation and discarded. The affinity gel was washed extensively with IP buffer, followed by repeated washing with IP buffer that lacked detergent, glycerol and protease inhibitors. Samples were eluted using 4 mg/mL 3xFLAG peptide at 4 °C for 30 min with agitation every 5 min[12]. Eluted proteins were then analyzed by silver stained SDS-PAGE and subjected to negative stain TEM.

## Immunoprecipitation - mass spectrometry analysis (IP-MS)

IPs from nuclear lysates of AT3 from human FU-UR-1 cells (eight replicates) and mouse AT3-initiated tumor tissues (eight replicates) using the anti-ASPSCR1 antibody were performed as above. Duplicate control IgG IPs were performed in parallel for both the FU-UR-1 cells and mouse tumor tissues. Proteins were eluted by boiling for 5 min in elution buffer (HEPES pH 8.0 50 mM, 2% SDS, DTT 5 mM). Eluted immunoprecipitates were incubated at 45 °C for 30 min, and then alkylated with 400 mM iodoacetamide for 30 min at 24 °C. Reactions were quenched with the addition of 200 mM dithiothreitol. Eluted proteins were prepared for trypsin digestion using the single-pot solid-phase-enhanced sample preparation (SP3) paramagnetic bead cleanup protocol as previously described[49]. Acetonitrile was added to the SP3 bead-protein mixture to a final 50% vol/vol, and incubated for 8 min at room temperature. Using a magnetic rack, beads were washed two times with 200 μL 70% ethanol for 30 s, and once with 180 μL 100% acetonitrile for 15 s. For digestion, beads were reconstituted in 5 μL 50 mM HEPES pH 8.0 buffer containing trypsin/rLys-C enzyme mix (Promega) at a 1:25 enzyme to protein ratio, and incubated for 14 h at 37 °C. Digested peptides were recovered by removing the supernatant on a magnetic rack. The peptides from each sample were labeled with individual Tandem Mass Tags (TMT) (Thermo Fisher Scientific) as previously described[50], then concentrated in a SpeedVac centrifuge to remove acetonitrile, combined, and desalted before MS analysis on an Orbitrap Fusion (ThermoFisher).

Data acquisition on the Orbitrap Fusion (control software version 2.1.1565.20) was carried out using a data-dependent method with multi-notch synchronous precursor selection MS3 scanning for TMT tags. Survey scans covering the mass range of 380–1500 were acquired at a resolution of 120,000, with quadrupole isolation enabled, an S-Lens RF Level of 60%, a maximum fill time of 50 ms, and an automatic gain control (AGC) target value of 5e5. For MS2 scan triggering, monoisotopic precursor selection was enabled, charge

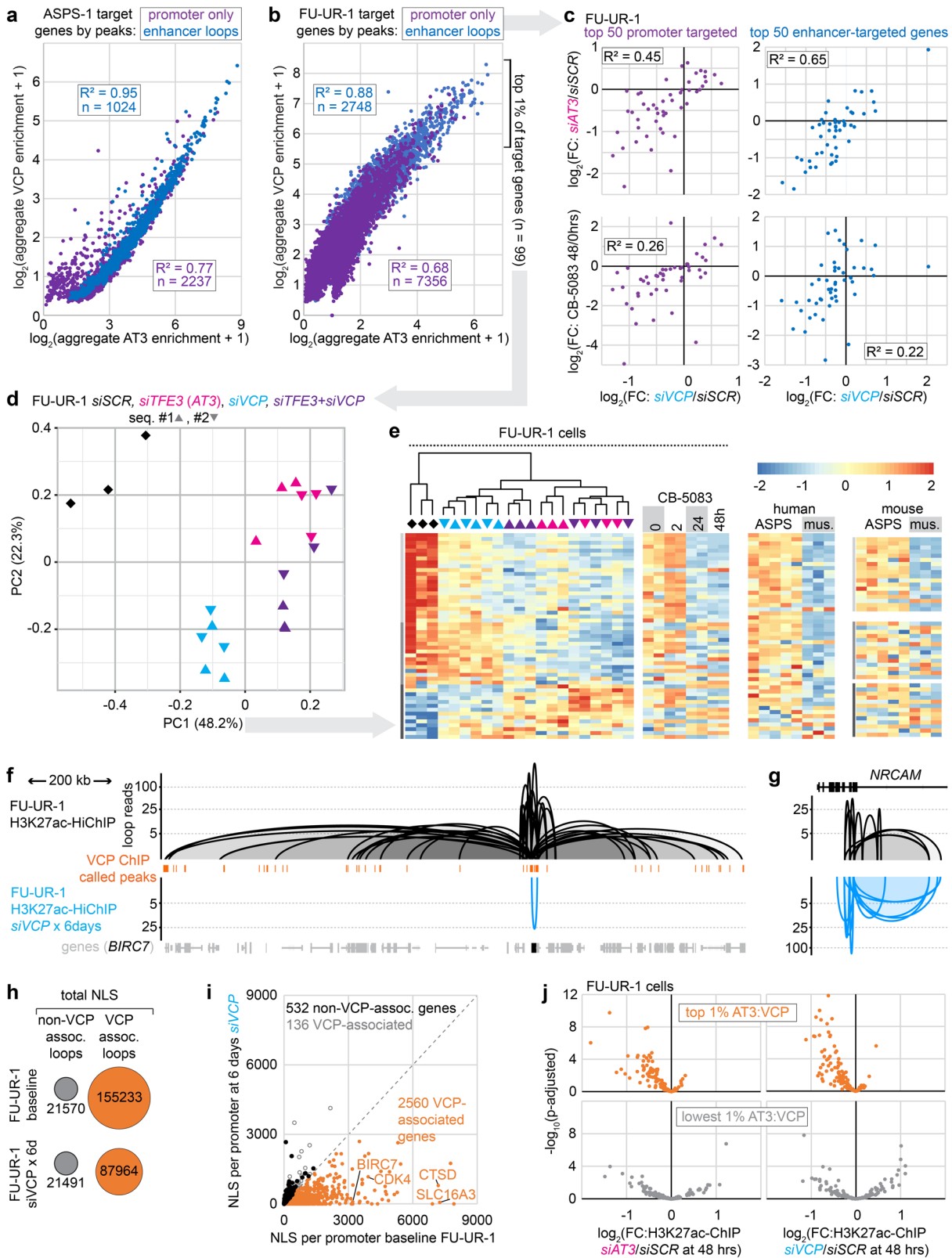

state filtering was limited to 2–4, an intensity threshold of 5e3 was employed, and dynamic exclusion of previously selected masses was enabled for 60 seconds with a tolerance of 20 ppm. MS2 scans were acquired in the ion trap in Rapid mode after CID fragmentation with a maximum injection time of 45 ms, quadrupole isolation, an isolation window of 1 $m/z$, collision energy of 35%, activation Q of 0.25, injection for all available parallelizable time turned OFF, and an AGC target value of 5e3. Fragment ions were selected for MS3 scans based on a precursor selection range of 400–1800 $m/z$, ion exclusion of 20 $m/z$ low and 5 $m/z$ high, and isobaric tag loss exclusion for TMT. MS3 scans were acquired in the Orbitrap after HCD fragmentation (NCE 65%) with a maximum injection time of 200 ms, 60,000 resolution, 120–750 $m/z$ scan range, ion injection for all parallelizable time turned OFF, and an AGC target value of 1e5. The total allowable

**Fig. 6 | Chromatin conformation enhancer loops depend on AT3:VCP.**
**a** Correlation of AT3 and VCP enrichments aggregated for each target gene in ASPS-1 cells from all narrowly defined peaks localized to the promoter or a looped enhancer (normalized by H3K27ac-HiChIP loop score). **b** Similar FU-UR-1 plot. **c** Correlation between log$_2$fold-change expression from siRNA VCP depletion over control in FU-UR-1 cells, with AT3 depletion or CB-5083 VCPi for the 50 strongest promoter-targeted genes and 50 strongest enhancer-targeted genes. **d** Principal component (PC) analysis according to the top one percent ($n = 99$, by AT3 aggregate enrichment) of genes by RNA-seq after *siSCR*, *siTFE3*, *siVCP*, or combinations in FU-UR-1 cells. **e** K-means cluster heatmaps of PC1-contributing genes (coefficients greater than 0.05, $n = 54$) in FU-UR-1 cells, showing expression after application of siRNAs or CB-5083 in FU-UR-1 cells as well as expression in human and mouse ASPS tumors compared to skeletal muscle. **f** Example tracks of H3K27ac-HiChIP loops around a highly targeted gene in baseline FU-UR-1 cells (above, black) and FU-UR-1 cells exposed to *siVCP* for 6 days (below, cyan; $n = 2$ siRNAs against VCP). **g** Negative control example tracks show retained/strengthened H3K27ac-HiChIP loops after filtering out VCP-ChIP-seq-peak-associated loops. **h** Comparison dot diagrams of the change in VCP-peak-associated or not-associated H3K27ac-HiChIP loops at baseline compared to 6 days of VCP depletion with siVCP ($n = 2$ for each). **i** Normalized loop score (NLS) aggregated per promoter for VCP-associated genes in FU-UR-1 cells at baseline (from 6b) or 6 days VCP depletion. Genes with no change or increased NLS per promoter are designated as VCP-associated or not. **j** Plot of differential H3K27ac ChIP-seq enrichment in FU-UR-1 cells with siRNA-depleted AT3 (left) or VCP (right) at loci defined as VCP-peaks and HiChIP loop ends that are top-ranked for AT3 enrichment as well (above) versus the lowest AT3 enriched HiChIP loop H3K27ac ChIP peaks that are not coincident with VCP peaks (below). ($n = 3$, biologically independent samples, Wald test $p$-values, Benjamini-Hochberg correction.).

cycle time was set to 4 s. MS1 and MS3 scans were acquired in profile mode, and MS2 in centroid format.

Raw MS data were processed using Sequest HT in Proteome Discoverer software (v2.4; Thermo Fisher Scientific), searching against the human (2020/02/08, Swiss-Prot) or mouse (2020/01/20, Swiss-Prot) UniProt reference proteomes plus a list of common contaminants. Precursor mass tolerance was set at 20 ppm and fragment mass tolerance at 0.8 Da. Dynamic modifications included Oxidation (+15.995 Da, M), Acetylation (+42.011 Da, N-Term), and static modification included Carbamidomethyl (+57.021 Da, C) and TMT (+229.163 Da, K, N-Term). Peptide identification FDR was calculated using Percolator by searching the results against a decoy sequence set; only peptide-to-spectrum matches (PSMs) with FDR < 1% were retained in the analysis. All PSM reporter ion values were corrected for isotopic impurities provided by the manufacturer.

Downstream data processing was performed in R software. PSMs that were assigned to a unique master protein and were not part of a common contaminant list were median aggregated into unique peptides. To calculate a protein fold enrichment of IP over control samples, PECA[51] analysis was performed at the peptide level, using the modified t-statistic parameter. Total protein enrichment was calculated as: $-\log_{10}$(FDR adjusted $p$-value) $\times$ (2 $\times$ fold-enrichment).

The mass spectrometry proteomics data have been deposited to the ProteomeXchange Consortium via the PRIDE[52] partner repository with the dataset identifier PXD022515.

## Lysis for western blot
Cells were washed and scraped in cold PBS and centrifuged at $500 \times g$ for 5 min at 4 °C. Cells were lysed in 1× RIPA buffer plus 1× protease inhibitor cocktail for 30 min on ice with mild shaking. Supernatant was collected after centrifugation at 13,000 rpm for 15 min at 4 °C.

## SDS-PAGE, Coomassie staining and mass spectrometry
Precast gels were used for protein electrophoresis. Protein samples were mixed with 2× Laemmli buffer, boiled for 4 min, spun and loaded in the gel. After electrophoresis, each gel was stained with 0.25% CBB (Coomassie Brilliant Blue R-250) for 1 h at RT with shaking. Each gel was destained with destaining solution (45%Methanol, 10% Glacial Acetic acid, 45% Water) till the protein bands were clearly visible. Gel images were captured on Azure Biosystems C300 and band of interest was identified by visual analysis. Each protein band was cut by a clean blade and sent to the proteomics core facility at University of Utah for mass spectrometry analysis to identify the protein in the band.

## Proximity ligation assay
Fresh frozen tumor specimens were embedded in optimal cutting temperature compound for cryostat sectioning at 5 μm thickness onto glass slides. The frozen sections were fixed in 10% formalin for 10 min then rinsed with water, followed by permeabilization with 0.1% Triton X-100 for 5 min. Sections were blocked with Duolink® Blocking Solution and incubated overnight at 4 °C with the indicated primary antibodies before performing proximity ligation using the Duolink® In Situ Red Starter Kit Mouse/Rabbit (MilliporeSigma) according to manufacturer's protocol. Briefly, primary antibodies were removed, and sections were washed once with PBST and twice with PBS followed by incubation with PLA probes for 20 min at room temperature. After two washes with Wash Buffer A from the Duolink® kit, sections were incubated with the ligation reagent at 37 °C for 30 min then washed twice with Wash Buffer A. Amplification reagent and Alexa fluor 488 anti-rabbit IgG (Thermo Fisher Scientific) were added to sections and incubated at 37 °C for 100 min in the dark, followed by two washes with Wash Buffer B, and one wash with 0.1% Wash Buffer B in PBS. Coverslips were mounted over sections with VECTASHIELD antifade mounting medium with DAPI. Fluorescence signals were visualized using a Zeiss Axiovert microscope with a 63× objective. Images were captured with z-stacks at 0.24 μm thickness and displayed at maximal intensity projection.

## Immunocytochemistry and confocal microscopy
For microscopy, 25,000 cells per chamber were seeded in a 4 chambered slide. After incubation, growth medium was removed and washed with PBS, fixed with 4% paraformaldehyde for 10 min, and washed three times with PBS for 5 min each. Cells were permeabilized with PBS containing 0.1% Triton X-100 for 10 min at room temperature and washed three times for 5 min. Cells were blocked with 3% BSA in PBST for 30 min at room temperature. Cells were incubated overnight at 4 °C in a humidified chamber with primary antibody diluted in blocking solution. Cells were washed three times 5 min each and incubated with fluorophore labeled secondary antibody for 1 h at room temperature in darkness. After incubation, cells were washed three times 5 min each. Coverslips were mounted with mounting media containing DAPI. The slide was then visualized by confocal microscopy.

## Western blotting
After SDS-PAGE, protein was transferred to PVDF and blocked in 5% milk for 2 h at room temperature. Membranes were incubated overnight at 4 °C in primary antibody diluted in blocking solution. Membrane were washed in TBST (0.2% Tween 20) three times 15 min each, then incubated with HRP conjugated secondary antibody at RT for 2 h and washed three times for 15 min each. ECL was added to the membrane, incubated for 2 min and then exposed to X-ray film and developed.

## Nuclear/cytoplasmic fractionation
Nuclear and cytoplasmic fractionation of cells was performed according to prior protocols[53,54] with some modifications. Briefly, cells were washed and scraped in cold PBS, centrifuged at $500 \times g$ for 5 min at 4 °C and washed with cold PBS. Cells were resuspended in Mitochondrial Isolation buffer (MIB; 250 mM Sucrose, 20 mM HEPES, 10 mM KCl, 1.5 mM MgCl$_2$, 1 mM EDTA, 1 mM EGTA, protease and

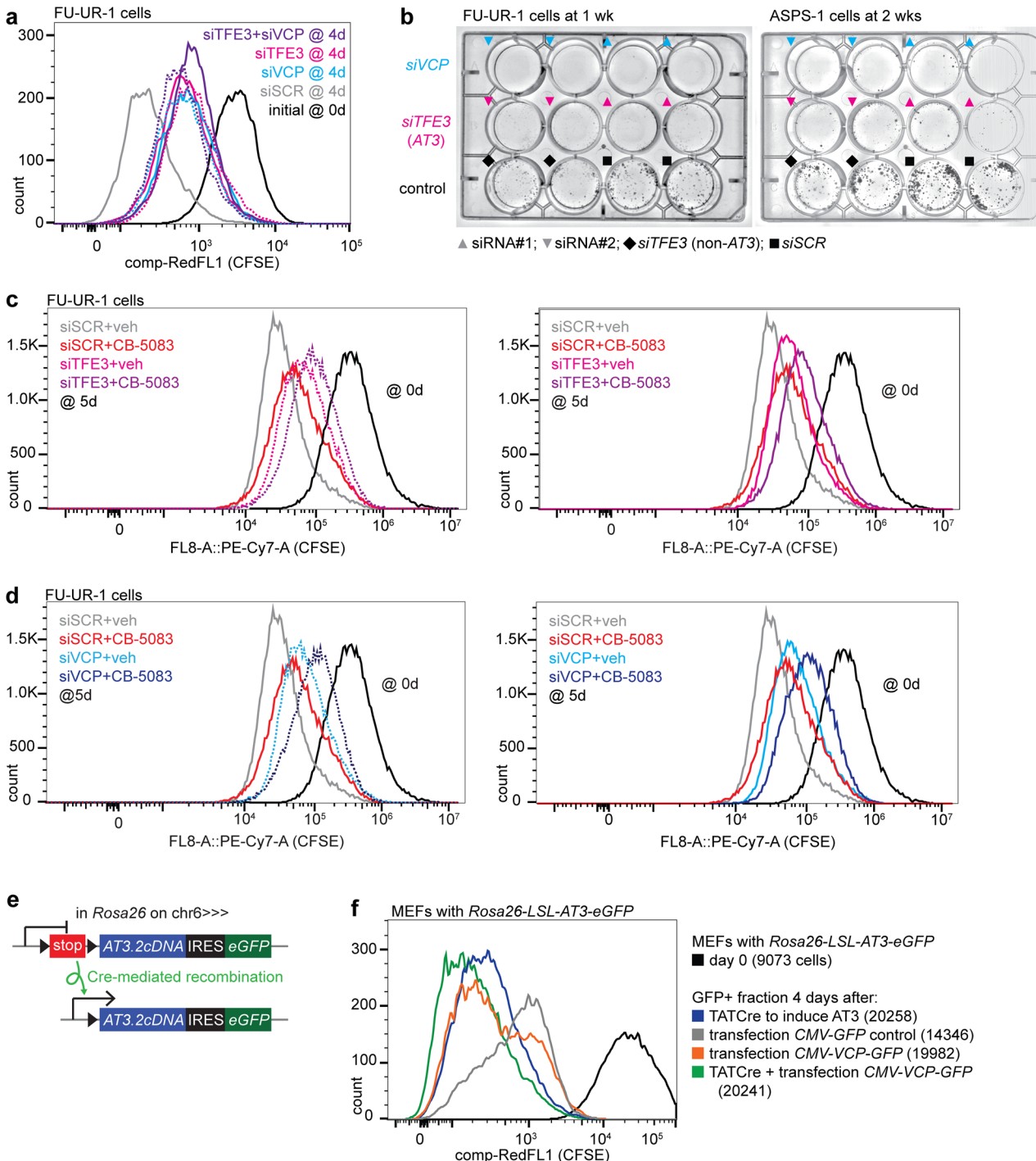

**Fig. 7 | AT3:VCP is a targetable functional dependency in cancer cells in vitro.**
**a** Flow-cytometry for CFSE dye depletion by proliferation after *siSCR*, *siTFE3*, *siVCP* and combinations in FU-UR-1 cells. (siRNA#1 for each is represented by a solid line; siRNA#2 for each is represented by a dotted line.) **b** Clonogenic assays after siRNAs applied to both the FU-UR-1 cell line and the ASPS-1 cell line (black diamonds represent an *siTFE3* that targets a portion of *TFE3* not included in *AT3*). **c** CFSE dye depletion assay combining siRNA depletion of AT3 by *siTFE3*#1 (solid line) and *siTFE3*#2 (dotted line) or control *siSCR* with CB-5083 or vehicle, showing that AT3 depletion or CB-5083 diminishes proliferation (AT3 depletion slightly more than CB-5083), but that their combination further diminishes proliferation (pushes distribution further to the right). (siRNA#1 for each is represented by a solid line; siRNA#2 for each is represented by a dotted line.) **d** The same for *siVCP*#1 (solid

line) and *siVCP*#2 (dotted line) or control *siSCR*, with CB-5083 or vehicle, showing the same phenomenon. **e** Schematic of the Rosa26-LSL-AT3-IRES-eGFP allele that is activated by Cre-recombinase-mediated excision of a stop sequence between the AT3 coding sequence and the native Rosa26 promoter. This allele, when activated in living mice led to the fully penetrant development of ASPS tumors. **f** Flow-cytometry for CFSE dye depletion by proliferation in GFP+ sorted mouse embryonic fibroblasts (MEFs) that bear an ASPSCR1-TFE3 allele that is induced by Cre-mediated recombination to remove a stop sequence between the promoter and the cDNA coding sequence. (Cells counted at day 0 were 9,073; ells counted at day 4 were 20,258 for TATCre added alone, 20,241 for TATCre and VCP-GFP added, 19,982 for VCP-GFP added alone, and 14,346 for GFP control added alone).

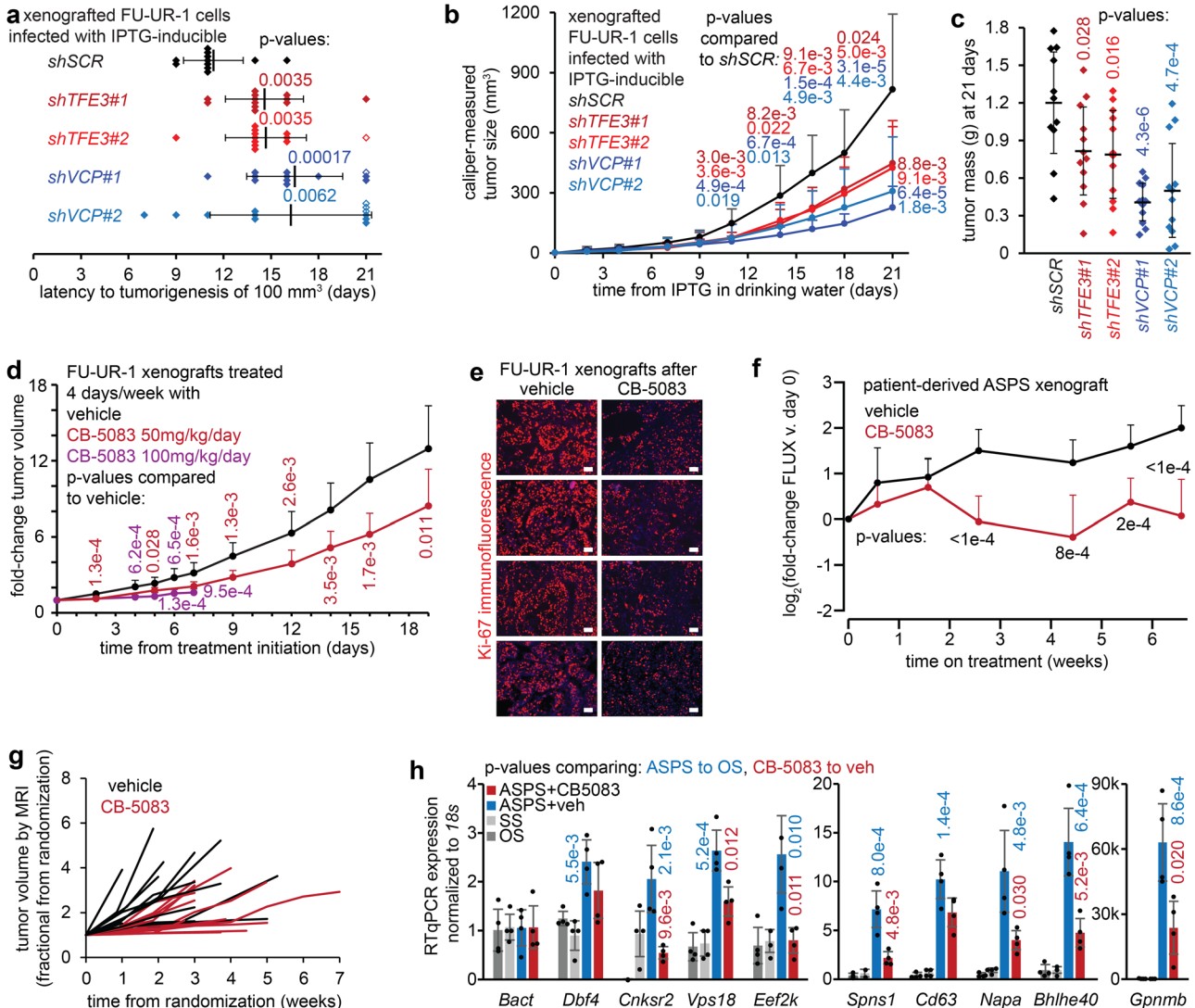

**Fig. 8 | AT3:VCP is a targetable functional dependency in cancer cells in vivo.**
**a** Time to tumorigenesis (100 mm³ calculated size) of FUUR-1 cells, transfected and selected for IPTG-inducible shRNAs, xenografted into NRG mice (1 million cells per flank in Matrigel). Drinking water IPTG began on day 0, three days after xenograft injection. (n = 4 for each transfected population of cells, in which each shRNA has n = 3. Mean ± standard deviation, two-tailed homoscedastic t-test p-values. Open diamonds represent tumors <100 mm³ at 21 days, counted that days for statistics). **b** Growth curves for the same transfected xenografts (n = 12 per shRNA, mean ± standard deviation, two-tailed homoscedastic t-test p-value). **c** Final mass for each transfected xenograft tumor harvested (day 21, mean ± standard deviation, two-tailed homoscedastic t-test p-value). **d** Growth of FU-UR-1 xenografts (NRG mice) treated with CB-5083 at 100 mg/kg/day for 4 days per week versus vehicle control (harvested day 8, due to gavage toxicity) or at 50 mg/kg/day for 4 days per week

versus vehicle control (n = 10 tumors per group; homoscedastic t-test p-values). **e** Immunofluorescence Ki-67 photomicrographs of FU-UR-1 xenografts harvested day 8 after treatment with 100 mg/kg CB-5083 or vehicle control, administered on days 1–4 and 8 (magnification bars, 50 μm). **f** Patient-derived xenograft growth data by luciferase activity on weekly measurements (n = 10, Tukey's test p-values) **g** Fractional growth curves for individual tumors in genetically induced mouse ASPSs, randomized to treatments at volumes >10 mm³ by serial MRI (n = 13, CB-5083; n = 15, vehicle). **h** RT-qPCR for target genes (defined as mouse AT3-ChIP-seq targeted and having reduced expression of homologues in both FU-UR-1 and ASPS-1 cells with AT3 depletion) in control tumors (OS osteosarcoma, SS synovial sarcoma) or genetically induced mouse ASPS tumors treated with vehicle or CB-5083 for 50 mg/kg for 4 days prior to harvest. (n = 4 tumors per group per gene; homoscedastic two-tailed t-test p-values).

phosphatase inhibitor cocktail) by pipetting and incubated for 10 min on ice. Homogenates were centrifuged at 1000 × g for 10 min at 4 °C. Supernatant fractions were recentrifuged twice at 16,000 × g for 20 min at 4 °C and supernatant-containing cytosolic fractions were collected. Pelleted nuclei were washed three times with ice cold hypotonic lysis buffer (HLB; 10 mM Tris-HCl, pH 7.5, 10 mM NaCl, 3 mM MgCl₂, 0.3% NP-40, Protease and phosphatase inhibitor cocktail) by 5 min incubation on ice, pipetting and vortexing, then spinning at 4 °C at 500 × g for 5 min. The nuclear pellet was resuspended in ice cold nuclear lysis buffer (NLB; 20 mM Tris-HCl, pH 7.5, 0.15 M NaCl, 3 mM MgCl₂, 0.3% NP-40, 10% Glycerol, Protease and phosphatase inhibitor cocktail). Nuclei were sonicated at 20% power 3X for 15 s in at 4 °C.

Lysate was centrifuged at 13,000 rpm for 15 min at 4 °C and supernatant collected as nuclear fraction and stored at −80 °C.

### Transfections
siRNAs were transfected into FU-UR-1 and ASPS-1 cells using Lipofectamine RNAiMAX (Invitrogen). To ensure robust protein reduction, siRNA and Lipofectamine reagent were increased by 25%. Per six centimeter plate, 22.5 μL Lipofectamine reagent in 300 μL Opti-MEM were combined with 7.5 μL siRNA (10 μM) in 300 μL Opti-MEM and incubated at room temperature for 5 min before adding to the media. Transfections were performed on two successive days with no media replacement before incubation for the specified duration.

For smaller transfections into HEK293T cells (Fig. 1 experiments) Lipofectamine3000 reagent was used according to the manufacturer's instructions.

For the larger scale transfections of plasmids into HEK293T cells, calcium phosphate was used as the transfection reagent. Per ten centimeter plate, 10 μg plasmid DNA in 500 μL double distilled water was thoroughly mixed with 50 μL 2.5 M CaCl$_2$. 500 μL 2X HBS pH 7.05 (280 mM NaCl, 10 mM KCl, 1.5 mM Na$_2$HPO$_4$H$_2$O, 50 mM HEPES) was added drop-wise and the mixture was incubated for thirty minutes at room temperature. Media was aspirated and DNA/CaPO$_4$ precipitates were gently applied drop by drop to the cells and incubated for 30 min at room temperature. Finally, media was gently added to the plates.

### Single cross-linking ChIP

For preparation of frozen tumor tissue, the tissue was placed in a Covaris tissueTUBE system and pulverized using a hammer and aluminum block on dry ice. The cellular powder was resuspended in PBS + protease inhibitors and cross linked in 1% formaldehyde for 10 min at room temperature. For preparation of cell cultures, cells were washed once with PBS and crosslinked in the dish using 1% formaldehyde in PBS + protease inhibitors for 10 min at room temperature. At this point, samples were treated similarly. Crosslinking was quenched with 125 mM glycine/PBS for 5 min at room temperature. Samples were washed three times with cold PBS + protease inhibitors. To isolate the nuclear fraction, 10 mL of cold Farnham cell lysis buffer (5 mM PIPES pH 8.0/85 mM KCl/0.5% NP-40) + PI was added to the cells and incubated on ice for 10 min. Cell membranes were further broken down by mechanical douncing and samples were passed through a 70 micron filter and centrifuged for 10 min at 4 °C and 1000 × $g$. The supernatant was dumped and the nuclear pellet was lysed with RIPA lysis buffer (1% NP-40/0.5% Na deoxycholate/1% SDS/PBS) + PI at a ratio of 1 mL per 300 mg of tissue. Samples were incubated on ice for 10 min, then diluted to 0.1% SDS and then proceeded to sonication.

Samples, divided into one ml aliquots in 1.5 ml sonication tubes, were sonicated for four cycles with 30 s intervals at 40% amplitude in freezer-chilled sonication blocks (ACTIVE MOTIF) to an average fragment size of 400–800 bp. Centrifugation cleared-off debris (12,000 rpm, 4 °C). After one hour pre-clearance with 30 μL per ml washed (0.5 mg/mL BSA/PBS) magnetic Dynabeads M-280 (Invitrogen) at 4 °C with rotation, the beads were disposed and samples were incubated over-night with five μg antibody at 4 °C with rotation. Centrifugation cleared-off debris (12,000 rpm, 4 °C) and IPed chromatin was combined with 100 μL washed bead slurry and incubated for 4.5-6 hours with rotation at 4 °C. The beads were collected on a magnetic stand and washed six times with ChIP RIPA Wash Buffer (10 mM Tris-HCl pH 7.5, 140 mM NaCl, 1 mM EDTA, 0.5 mM EGTA, 0.5% (v/v) NP-40, 0.5% (v/v) Triton X-100, 0.1% SDS, 0.1% NaDOC), twice with ChIP LiCl Wash Buffer (10 mM Tris-HCl pH 8.0, 1 mM EDTA, 500 mM LiCl, 0.5% (v/v) NP-40, 0.5% NaDOC), and once with TE (10 mM Tris-HCl pH 8.0, 1 mM EDTA).

Beads were incubated with Proteinase K (Qiagen) (40 μg/100 μL ChIP Elution Buffer [10 mM Tris-HCl pH 8.0, 150 mM NaCl, 5 mM EDTA, 1% SDS]) for 60 min at 37 °C followed by 270 min incubation at 65 °C. The beads were discarded and chromatin was cleaned-up using DNA Clean & Concentrator-5 (Zymo Research).

### Native ChIP-seq

Native ChIP was performed as described[55].

### Double cross-linking ChIP

Cultured cells were washed three times with HBSS and cross-linked with fresh 20 mM dimethyl pimelimidate (dissolved in 0.2 M Triethanolamine pH 8.2) for 60 min at room temperature with rotation. For the final 10 min, formaldehyde was added (1% final concentration) before quenching with glycine (125 mM final) followed by three washes with cold PBS. Cells were lysed in Mild Cell Lysis Buffer (10 mM Tris-HCl pH 8.5, 10 mM NaCl, 0.5% (v/v) NP-40) supplemented with 1× protease inhibitor cocktail (PIs) for ten minutes on ice. Pelleted nuclei (1200 rpm) were washed once with Nuclei Wash Buffer (10 mM Tris-HCl pH 8.5, 200 mM NaCl, 1 mM EDTA, PIs).

Tumors were dissociated into single cell suspensions using Tumor Dissociation Kit (Miltenyi Biotec). Cells were washed, cross-linked, re-washed and lysed as for cultured cells. Next, each sample was dounced on ice by twenty compressions with a loose dounce followed by ten compressions with a tight dounce before filtration through a 40 μm nylon mesh cell strainer. Nuclei were washed similarly to cultured cells.

Nuclei were resuspended in Nuclei Lysis Buffer (50 mM Tris-HCl pH 8.0, 10 mM EDTA, freshly added 1% SDS, PIs) by brief pipetting and immediate dilution 1:10 in ChIP Dilution Buffer (16.7 mM Tris-HCl pH 8.1, 16.7 mM NaCl, 1.2 mM EDTA, 1.1% Triton X-100, freshly added 0.01% SDS, PIs) followed by ten minutes incubation on ice. Samples, divided into one mL aliquots in 1.5 mL sonication tubes, were sonicated for four cycles with 30 s intervals at 40% amplitude in freezer-chilled sonication blocks (Active Motif) to an average fragment size of 400–800 bp. Centrifugation cleared-off debris (12,000 rpm, 4 °C). After one hour pre-clearance with 30 μL per mL washed (0.5 mg/mL BSA/PBS) magnetic Dynabeads M-280 (invitrogen) at 4 °C with rotation, the beads were disposed and samples were incubated overnight with five μg antibody at 4 °C with rotation. Centrifugation cleared-off debris (12,000 rpm, 4 °C) and IPed chromatin was combined with 100 μL washed bead slurry and incubated for 4.5–6 h with rotation at 4 °C. The beads were collected on a magnetic stand and washed six times with ChIP RIPA Wash Buffer II (10 mM Tris-HCl pH 7.5, 140 mM NaCl, 1 mM EDTA, 0.5 mM EGTA, 0.5% (v/v) NP-40, 0.5% (v/v) Triton X-100, 0.1% SDS, 0.1% NaDOC), twice with ChIP LiCl Wash Buffer (10 mM Tris-HCl pH 8.0, 1 mM EDTA, 500 mM LiCl, 0.5% (v/v) NP-40, 0.5% NaDOC), and once with TE (10 mM Tris-HCl pH 8.0, 1 mM EDTA). Beads were incubated with Proteinase K (Qiagen) (40 μg per 100 μL ChIP Elution Buffer [10 mM Tris-HCl pH 8.0, 150 mM NaCl, 5 mM EDTA, 1% SDS]) for 60 min at 37 °C followed by 270 min incubation at 65 °C. The beads were discarded and chromatin in was cleaned-up using DNA Clean & Concentrator-5 (Zymo Research).

### H3K27ac ChIP after knock-down

Cells were washed once with room temperature PBS, cross-linked with 1% formaldehyde (10 min), quenched with 125 mM glycine, and washed three times with cold PBS. Cell pellets were lysed with cold Cytoplasmic lysis Buffer (20 mM Tris-HCl pH 8.0, 85 mM KCl, 0.5% (v/v) NP-40) supplemented with 1× protease inhibitor cocktail (PIs) for 5 min on ice. Nuclear pellets (1200 rpm) were resuspended in SDS Lysis Buffer (50 mM Tris-HCl pH 8.1, 10 mM EDTA, 1% SDS, PIs) and incubated on ice for 10 min. Samples were sonicated for four cycles with 30 s intervals at 40% amplitude in freezer-chilled sonication blocks (ACTIVE MOTIF) to an average fragment size of 200–400 bp. Centrifugation was performed to clear-off debris (12,000 rpm, 4 °C). Sonicates were diluted 1:10 in ChIP Dilution Buffer (16.7 mM Tris-HCl pH 8.1, 167 mM NaCl, 1.2 mM EDTA, 0.25% Triton X-100, 0.01% SDS, PIs). After one hour pre-clearance with 30 μL per mL washed (0.5 mg/mL BSA/PBS) magnetic Dynabeads M-280 (invitrogen) at 4 °C with rotation, the beads were disposed and samples were incubated overnight with five μg antibody (abcam ab4729) at 4 °C with rotation. Centrifugation cleared-off debris (12,000 rpm, 4 °C) and IPed chromatin was combined with 100 μL washed bead slurry for five hours incubation with rotation at 4 °C. The beads were collected on a magnetic stand and washed three times with ChIP RIPA Wash Buffer I (10 mM Tris-HCl pH 8.0, 140 mM NaCl, 1 mM EDTA, 1% (v/v) Triton X-100, 0.1% SDS, 0.1% NaDOC), three times with ChIP RIPA High Salt Wash Buffer (10 mM Tris-HCl pH 8.0, 360 mM NaCl, 1 mM EDTA, 1% (v/v) Triton X-100, 0.1% SDS, 0.1% NaDOC), twice with ChIP LiCl Wash Buffer (10 mM Tris-HCl pH 8.0, 1 mM EDTA, 250 mM LiCl, 0.5% (v/v) NP-40, 0.5% NaDOC), and once with TE (10 mM Tris-HCl

pH 8.0, 1 mM EDTA). Beads were resuspended in 100 µL Low SDS ChIP Elution Buffer (10 mM Tris-HCl pH 8.0, 300 mM NaCl, 10 mM EDTA, 0.1% SDS, fresh 5 mM DTT) and incubated for 45 min at 65 °C before addition of 40 µg Proteinase K (Qiagen) and further incubation for 1.5 h at 37 °C. The beads were discarded and chromatin was cleaned-up using DNA Clean & Concentrator-5 (Zymo Research).

### ChIP-seq

Libraries were constructed with the NEBNext ChIP-Seq Library Prep with Unique Dual Index Primers (New England Biolabs) and sequenced on an Illumina NovaSeq instrument using the 2 x 50 bp protocol to a depth of approximately 30 M reads per sample. Reads were aligned to mm10 mouse genome version for mouse samples and hg38 human genome version for human samples using BWA-MEM (version 0.7.10-r789) for paired-end reads[56].

Peaks were called from each of the aligned bam files against input reads using MACS2[57], version 2.2.6 with the parameters: callpeak -B --SPMR --*p*value = 1e-10 --mfold 15 100. ChIP input was used as the background for MACS2. MACS was used to produce normalized bedgraphs, which were subsequently converted to bigWig files. Peaks were filtered to remove peaks that are in blacklist, including ENCODE blacklisted regions[58]. Duplicate reads were removed using samtools rmdup for all downstream analyses[59]. Merged bigWig enrichment files for each condition with multiple replicas were generated in an average manner followed by normalization of read depth. We used --broad tag to call broad peaks for histone modification ChIP-seq data.

Highly Enriched regions were called by a modification of ROSE, which emulates the original strategy used by the Young lab to call superenhancers[60,61]. Briefly, peaks are found in a normal mode. Then, peaks within a given distance (this was set at 40 kb) are 'stitched' together into larger regions. The enrichment signal of each of these regions is determined by the total number of reads subtracting the input reads, each over the genomic distance of the stitched peaks, ignoring the inter-peak distances. After plotting this signal normalized to the highest region set at a value of one against the fractional ranking of each AT3 enhancer element (with the highest rank set at 1), every cluster of AT3 peaks on this curve above the inflection point of slope equals one was designated a highly enriched region or cluster of AT3 peaks. Next all genes annotated as nearest genes for peaks in the region were identified, to reflect the likely broad regulation region.

Differential Peak analysis for H3K27ac enrichment changes after knock-down of AT3 or VCP was performed using DESeq2 with the parameters "--min 100 --threshold 0.001". Peaks called from H3K27ac ChIP-seq were used as target features and they were scored based on the bigWig file. Lastly, peaks with minimum count of 100 and *q*-value 0.001 were considered as significantly changed peaks.

Motifs were called based on the 100 bp surrounding the summit. Motifs were discovered using MEME[62].

Heatmaps and profile plots for scores associated with genomic regions were generated with plotHeatmap following calculating scores per genome regions and prepared as an intermediate file with computeMatrix of deepTools 3.3.2[63].

The profile plots of peak distributions were generated using ChIPpeakAnno (version 3.22.0).

### RNA purification, cDNAs synthesis, and qPCR

Cells were lysed in Trizol Reagent (Ambion) and total RNA was purified with Direct-zol RNA Miniprep Plus Kit (Zymo Research) including DNase I treatment. cDNA was synthesized using High Capacity cDNA Reverse Transcription Kit (Applied Biosystems) and amplified in triplicate on the QuantStudio 12 K Flex system with PowerSYBR Green PCR Master Mix (Applied Biosystems). Differential gene expression for each target (primers listed in Supplementary Table 3) was calculated by the comparative ΔΔCt method using *HPRT1* or *ACTB* as a normalization control.

### RNA-seq

Libraries were constructed with the TruSeq Stranded mRNA Library Prep with Unique Dual Index Primers (Illumina) and sequenced on an Illumina NovaSeq instrument using the 2 × 50 bp protocol to a depth of approximately 25 M reads per sample. Reads were aligned using BWA-MEM (version 0.7.10-r789) for paired-end reads[56] to hg38 human genome version.

Count matrices were generated with featureCounts (version 1.6.3)[64] and differential expression analysis was performed with DESeq2 3.11[65]. Differentially expressed genes with a false-discovery rate of <0.05 were regarded statistically significant. The regularized log (rlog) counts were used for sample visualizations. PCA plots were generated using the first two principal components from the rlog values of the noted gene set. Heatmaps of this distance matrix were generated by unsupervised hierarchical clustering based on the sample Euclidean distance.

### Pathways analysis

The Enrichr program was used to test gene set enrichment analysis for the Reactome pathways[66,67]. The program identified significant pathways in both ASPS-1 and FU-UR-1 siTFE3-downregulated and highly enriched region annotated genes.

### HiChIP

HiChIP was performed according to a published protocol[22]. Libraries were constructed with HiChIP Library Prep kit with Nextera Unique Dual Indexes (Illumina). Sequencing was performed on an Illumina NovaSeq instrument using the NovaSeq Reagent Kit v1.5 150 × 150 bp protocol to a depth of 80 M reads per sample.

The HiChIP sequencing reads were aligned to the hg38 reference genome using the HiC-Pro pipeline[68]. Chromatin loops were identified by the Hichipper pipeline[69]. The loops with at least four pairs of valid reads (PETs) and an FDR value < 0.01 were kept as high-confidence loops for subsequent analyses.

For VCP-associated loops analysis, we used H3K27ac peaks as basic anchors; those anchors that are not intersected with VCP peaks are defined as control anchors. Loops that intersect with these anchors are defined as control loops. Anchors that intersect with VCP are referred as VCP anchors, and loops associated with them are referred as VCP loops. Similarly, VCP super peaks are used to define VCP SE loops. The bedtools intersection function is used for peaks intersection and pairToBed function is used to link loops with different types of anchors.

### Native co-immunoprecipitation

HEK293T transfected cells were lysed in Lysis Buffer (1 mL:6 g) (50 mM HEPES-KOH pH 6.8, 150 mM KOAc, 2 mM Mg(OAc)2, 1 mM CaCl2, 200 mM sorbitol) supplemented with 1× protease inhibitor cocktail (PIs) followed by centrifugation to clear-off debris (12,000 rpm, 4 °C). Anti-FLAG M2 Magnetic Beads (100 µL, SIGMA, M8823) were washed twice with ten packed bead volumes of TBS (50 mM Tris-HCl pH 7.4, 150 mM NaCl) and coupled with the protein suspension. Final volume was adjusted to one mL with Lysis Buffer and slurry was incubated over-night at 4 °C with rotation. The beads were collected on a magnetic stand and washed three times with twenty packed bead volumes of TBS. Protein complexes were eluted by two rounds of thirty minute incubations on ice with 200 ng per µL 3X FLAG Peptide (SIGMA, F4799) in TBS.

### High molecular weight native co-immunoprecipitation

HEK293T transfected cells were lysed in Lysis Buffer (50 mM Tris-HCl pH 8.0, 150 mM NaCl, 1 mM EDTA, 0.1% NP-40, freshly added 1 mM DTT) supplemented with 1× protease inhibitor cocktail (PIs). Lysate was dounced on ice by twenty-thirty compressions with a tight dounce followed centrifugation to clear-off debris (18,000 × *g*, 15 min, 4 °C).

Anti-FLAG M2 Magnetic Beads (100 μL, SIGMA, M8823) were washed three times with PBST (PBS − 0.1% Tween-20), coupled with the protein suspension, and incubated over-night at 4 °C with rotation. The beads were collected on a magnetic stand and washed three times with IPP150 Buffer (10 mM Tris-HCl pH 8.0, 150 mM NaCl, 1 mM EDTA, 0.1% NP-40, PIs) and three times with FLAG Native Elution Buffer (20 mM Bis-Tris pH 7.0, 20 mM NaCl, 1 mM EDTA, 0.02% NP-40, 200 mM ε-aminocaproic acid, PIs). Protein complexes were eluted by two rounds of thirty minute incubations on ice with 200 ng per μL 3X FLAG Peptide (SIGMA, F4799) in Native Elution Buffer.

### Blue Native PAGE

Cells were lysed in NativePAGE 1× Sample Buffer (Novex, Thermo-Fisher) supplemented with 1× protease inhibitor cocktail. When mentioned, the NativePAGE 1× Sample Buffer was supplemented with 1% DDM (n-dodecyl-β-D-maltoside) or Digitonin. G-250 Sample Additive was added to whole cell lysates or native immunoprecipitated samples for a final concentration of 0.005%. Immunoprecipitated samples were not supplemented with DDM or Digitonin. Samples were loaded on precast NativePage 4-16% Bis-Tris Gels (Novex, ThermoFisher). Electrophoresis and transfer to PVDF membrane were done according to the manual, followed by standard western blotting procedures.

### Production and purification of recombinant ASPSCR1 variants and VCP

pET16b plasmid encoding N-terminally His-tagged full length VCP, ASPSCR1, trASPSCR1$_{1-311}$ and full-length AT3 were produced in the *Escherichia coli* host strain BL21-CodonPlus(DE3)-RIL competent cells. 10 mL bacteria cells were grown overnight at 37 °C in LB media then transferred into 2 L ZY auto-induction media[45] and grown to optical density of 0.3 at 37 °C and then transferred to 19 °C for around 15 h. 1 mM Isopropyl β-D-Thiogalactoside (IPTG) was added into the cell culture and the cells were grown for 3 h at 19 °C. Cell pellets were harvested and lysed with Ni-A buffer (50 mM Tris (pH 7.5), 500 mM NaCl, 20 mM imidazole, 0.5 mM DTT, 20 mM MgCl$_2$) containing 25 μg/mL PMSF, 0.5 μg/mL aprotinin, 0.7 μg/mL pepstatin, 0.5 μg/mL leupeptin, 100 μg/mL lysozyme and 1 mg/mL DNase. Cell lysates were incubated on ice for an hour followed by sonication at 80% power for 2 × 2 min cycles. Sonicated lysates were centrifuged at 41,000 × $g$ for 45 min. The supernatants of the lysates were poured into equilibrated Ni-NTA resin. Eluted proteins were loaded onto either HiTrap-Q HP anion exchange columns (VCP, WT-ASPSCR1 and trASPSCR1$_{1-311}$) or HiTrap Heparin HP affinity columns (ASPSCR1-TFE3) followed by Superose 6 increase 10/300 GL sizing columns[14]. Proteins were pooled from gel filtration peaks and were analyzed by Coomassie Blue stained SDS-PAGE.

### Reconstitution of VCP-ASPSCR1 variant complexes

Purified VCP was incubated with different ASPSCR1 variants on ice for 4 h. The reconstituted complexes were subjected to Superose 6 increase 10/300 GL sizing columns and eluted samples were analyzed by Coomassie Blue stained SDS-PAGE.

### Transmission electron microscopy

3.5 μL of samples were applied to the surface of freshly glow-discharged continuous carbon grids. Excess sample was removed by blotting, followed by a brief washing step in water, and then blotted again to near dryness. Grids were stained with 1% uranyl acetate for 20 s, blotted to dryness and then air dried. Grids were imaged using either a Hitachi 7100 transmission electron microscope (TEM) operated at 70 kV equipped with a Gatan Orius CCD camera (20,000× magnification, images for AT3 and WT ASPSCR1 IP from HEK cells) or using a JEOL JEM-1400 TEM equipped with Gatan Orius CCD camera operated at 120 kV (30,000x magnification, 2D class averages for AT3 IP from HEK cells).

### 2D classification

For AT3 co-IP elution samples from HEK cells, a total of 64 negative stain micrographs were imported into RELION[70]. The pixel size of each micrograph was 2.2 Å/pixel. Particles were automatically selected using the Laplacian-of-Gaussian option within RELION and extracted with a box size of 138 pixels. Reference-free 2D class averaging was performed using a circular mask diameter of 91 pixels (Regularization parameter = 4). Subset selections were then performed for side views and top views. Total of 2906 particles for side views and 16,637 particles for top views were selected. 2D classifications were then performed again using selected particles. A total of 182 particles and 6394 particles were classified into the final side views and top views, respectively (Fig. 2D).

Images of the AT3/VCP co-sized sample were processed similarly. In brief, 64 micrographs at 2.5 Å/pixel were imported into RELION. Particles were also selected using the Laplacian-of-Gaussian option and 2D classification revealed 2301 and 1406 particles classified into side views and top views, respectively (Fig. S2D).

### ATPase assay

Complexes were reconstituted by combining 6 μM of recombinantly purified VCP with 6 μM of each ASPSCR1 variant on ice for 30 min. 40 μL of each complex were added into separate wells in 96-well plate. 10 μL 5 mM ATP were added into each reaction to initiate the reaction. The reactions were incubated at 37 °C for 10 min, followed by addition of 50 μL of malachite green. The reactions were quenched by 21% citric acid. Final concentrations of ATP and VCP in each reaction were 333 μM and 0.8 μM, respectively. Absorbance at 650 nm for each sample well was measured using a Synergy Neo plate reader (Bio Tek)[71]. The percentage of ATPase activity was calculated from the ratio of $\Delta A_{650}$ (between VCP-ASPSCR1 variant complex and buffer well) and $\Delta A_{650}$ (between VCP and buffer well). ATPase activity of recombinantly purified VCP was set to 100% and each reaction was measured four times.

### shRNA knock-down experiments

FUUR-1 cells were transduced with lentiviral Sigma MISSION Inducible shRNA Vectors, pLKO-puro-IPTG-3xLacO, expressing two different shRNAs targeting *TFE3*, or two different shRNAs targeting *VCP*, or non-targeting control shRNA, which sequences are identical to siRNA sequences described above. Three multiplicities of infection (viral particles to cells) in the presence of 8 μg/mL polybrene were applied to sub-confluent cells for 48 h. Media was replaced, and puromycin (25 μg/mL) selection began a day later to generate pools of cells derived from multiple transduction events. The selection continued until non-transduced control cells were completely eliminated. For shRNA induction, cells were treated with 1 mM IPTG for the indicated duration with IPTG refreshment every three days. For IPTG-inducible xenograft knockdown studies, drinking water containing 10 mM IPTG was replenished every 72 h. Drinking water included 1% glucose to overcome taste aversion.

### In vivo drug treatments

For the in vivo experiments, three models were used. The FU-UR-1 xenograft was previously described[72,73]. Briefly, NRG mice were the hosts of flank injected cells, $1 \times 10^6$ in matrigel. Tumor size was measured at least thrice weekly by calipers. After tumors reached a calculated size of 100 mm$^3$, mice were randomized to treatment groups. Mice were also weighed at least thrice weekly to assess for the gut-related toxicity that was anticipated. A body mass reduction of 20% was considered sufficiently morbid to stop treatments and euthanize the animal, humanely. The tumors harvested on day 8 from the control and 100 mg/kg CB-5083 treatment groups were processed for immunofluorescence according to the previously published protocol[74].

The ASPS PDX-derived cell line was developed at the University of Calgary. Briefly, under IRB approval (HREBA.CC-16-0144) and written informed consent, a clinical biopsy was collected from a 14-year old

female with ASPS (with AT3.2 fusion) undergoing surgery for a soft-tissue metastasis in the breast. Fresh tumor tissue from the mastectomy was sectioned into $1 \times 1 \times 1$ mm cubes and implanted into the flanks of NOD/SCID mice to establish a patient-derived xenograft (PDX). Established PDX tissue was removed from the 4th in vivo passage, dissociated by trituration, filtered using a 0.7 µm filter, washed and cultured in Opti-MEM medium supplemented with 10% heat-inactivated fetal bovine serum (FBS), 1% penicillin/streptomycin, and 50 µM 2-Mercaptoethanol at 37 °C in a humidified 5% $CO_2$ incubator.

This ASPS PDX-derived cell line was transduced with a lentiviral construct containing an mCherry fluorescent protein and firefly luciferase (pLV430G). Six-eight week-old female NOD.CB17-$Prkdc^{scid}$/NCrCrl (Charles River, Strain #394) were used in this study. Low passage $1 \times 10^6$ ASPS$^{mC/FLUC}$ cells were injected in the gastrocnemius muscle. Sixty days following implantation, tumor burden was evaluated by bioluminescence imaging using intraperitoneal injections of luciferin (GoldBio) and the Xenogen IVIS system. Animals were treated with 80 mg/kg CB-5083 for two weeks (4 days on, 3 days off), followed by a 1 week drug holiday, and resumption of treatment at 60 mg/kg CB-5083 for four weeks (4 days on, 3 days off). Luciferase imaging was performed weekly and all images processed with LivingImage Software (v4.7.3). Body weight and score was assessed every 2–3 days.

Mice heterozygous for expression of $CreER$ from the $Rosa26$ locus induced the floxed-stop conditional $AT3.2$ at the other $Rosa26$ allele by spontaneous "leaky" nuclear entry in the absence of administered tamoxifen. These mice began weekly imaging of the brain under general anaesthesia, using an HT Bruker BioSpec 7.1 T horizontal-bore MRI instrument. MRI tumor volumes were quantitated using ImageJ software by an operator blinded to mouse identity and treatment. When a tumor was identified that passed the entry criteria of volume greater than or equal to 10 mm³, the mouse was randomized to receive 100 mg/kg CB-5083 delivered daily on four consecutive days each week by oral gavage in 0.5% hydroxypropyl methylcellulose or this vehicle alone at equal volumes.

### Reporting summary

Further information on research design is available in the Nature Portfolio Reporting Summary linked to this article.

## Data availability

The genomics data generated in this study have been deposited in the Gene Expression Omnibus database under accession code GSE162609. The raw data genomics data from one patient are protected and are not available due to privacy laws, but processed data are included in the GEO deposit, along with all other raw and processed data. The proteomics data generated in this study have been deposited on the PRIDE database under accession code PXD022515. Source data for Figs. 4a, f–h, 5a, b, d–h, 6a–c, i, j, Supplementary Figs. 4a, 5b, d, f, and i are included in Supplementary Information/Source Data files. Source data are provided with this paper.

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

## Acknowledgements

This work was supported by NIGMS (NIH) grant R35GM133772 to P.S.S. Work at the University of British Columbia by J.S.E.Y., G.N., G.M., M.H., and T.O.N. was supported by grants from the Terry Fox Research Institute (1082) and the Canadian Cancer Society (705615). Work was supported by the L.B. and Olive S. Young Presidential Chair for Cancer Research, the Huntsman Cancer Foundation, the University of Utah Department of Orthopaedics and R01CA201396, U54CA231652 and 2P30CA042014-31 from the National Cancer Institute (NIH), all to K.B.J. We thank David Belnap and the University of Utah Electron Microscopy Core for TEM support, Brian Dalley and the High-Throughput Genomics Core at Huntsman Cancer Institute for sequencing support, Tim Parnell and the Bioinformatics Core at Huntsman Cancer Institute for sequencing alignments, Kate Modzelewska, Guoying Wang, and David Lum in the Preclinical Research Shared Resource at Huntsman Cancer Institute for cell line xenograft expertise, and the University of Utah Center for High Performance Computing for computational support.

## Author contributions

Conceptualization, K.J.; Methodology, P.S., G.M., M.H., J.V., J.G., X.Z., A.L.; Formal Analysis, A.P., L.L., S.W., S.C., C.H., G.N., K.M.; Investigation, S.V., A.P., S.W., J.B., B.J., S.K., J.Y., J.Z., M.N., S.L., L.C., K.S.-F., M.S., C.J., S.S., H.L., N.D., J.B., J.L.; Resources, A.L., T.N., J.G., R.W., D.S.; Writing—Original Draft, K.J.; Writing—Review and Editing, all; Visualization Preparation, A.P., S.W., L.L., K.J.; Supervision, Project Administration, and Funding Acquisition, D.M., F.Z., M.H., T.N., G.M., P.S., K.J.

## Competing interests

The authors declare no competing interests.

## Additional information

[1]Department of Orthopaedics, University of Utah, Salt Lake City, UT, USA. [2]Department of Oncological Sciences, University of Utah, Salt Lake City, UT, USA. [3]Huntsman Cancer Institute, University of Utah, Salt Lake City, UT, USA. [4]Department of Biochemistry, University of Utah, Salt Lake City, UT, USA. [5]Department of Pathology, University of British Columbia, Vancouver, BC, Canada. [6]Canada's Michael Smith Genome Sciences Centre, BC Cancer, Vancouver, BC, Canada. [7]Department of Microbiology, Immunology and Infectious Disease, University of Calgary, Calgary, AB, Canada. [8]Department of Oncology, Charbonneau Cancer Institute, Cumming School of Medicine, University of Calgary, Calgary, AB, Canada. [9]Department of Orthopaedic Surgery, University of California San Francisco, San Francisco, CA, USA. [10]Department of Oncology, McGill University and Lady Davis Institute for Medical Research, Montreal, QC, Canada. [11]Departments of Anatomic Pathology, Translational Molecular Pathology and Genomic Medicine, The University of Texas MD Anderson Cancer Center, Houston, TX, USA. [12]Department of Microbiology and Immunology, Michael Smith Laboratories, University of British Columbia, Vancouver, BC, Canada. [13]Department of Medical Genetics, University of British Columbia, Vancouver, BC, Canada. [14]These authors contributed equally: Amir Pozner, Li Li, Shiv Prakash Verma. ✉e-mail: kevin.jones@hci.utah.edu

