## [Peer Review File · Nature Communications]

ASPSCR1-TFE3 reprograms transcription by organizing enhancer loops around hexameric VCP/p97REVIEWER COMMENTS

Reviewer #1 (Remarks to the Author):

This is a solid study. There are a few points that would strengthen the article. In particular, it was not clear to this reader to what extent the inhibitions that were tried, whether by drug or siRNA, were specific to this tumor, do they affect other tumors, or do they affect all cells. For the sake of the reader, there are a few points throughout the manuscript that would benefit from clarification. For example:

What is a cofactor? In the introduction it was written: "We demonstrate that VCP is a likely obligate co-factor of ASPSCR1-TFE3, one of the only such fusion oncoprotein co-factors identified in cancer biology."

When would you consider something a co-factor for an oncoprotein? For example, some variants of ApoE are important for allowing metastasis. Are they co-factors? I am not objecting to the use of the term, but it would help frame the questions you are addressing to define it better. For example, to be a co-factor does it have to engage directly with the oncoprotein? Since this could be a useful concept for other cancers, a clear definition would be helpful.

What makes an oncoprotein "not readily targetable"

There are many reasons why different oncoproteins are not targetable. It might be useful here to list some of the different reasons and then, for those who do not work on AT3, what are the specific reasons why AT3 not targetable? It would help the reader to determine what can be generalized from this work.

On page 3, line 74 it is written: "Here, we identify VCP as the most important nuclear interactor with the fusion oncoprotein AT3."

It would help to say, What are the criteria for saying it the most important. Does it bind the most? Please clarify.

On page 7, lines 130-133:

"Co-transfection of C. with AT3 reduced the presence of higher molecular weight assemblies

in FLAG-AT3-IP (Fig. 2g). LAG-AT3-IP without co-transfected C. recovered assemblies as large as 1050 kD, consistent with a VCP hexamer complexed with six AT3 molecules.”

Did the transfection of C Δ disassemble the AT3 in the nucleus? Did C Δ need a nuclear localization signal to have an effect? Did it affect the cell physiology?

On page 12, line 265-266:

“AT3 depletion correlated better with VCP depletion for transcriptional changes among enhancer-targeted genes than among promoter-targeted genes.”

Is there any indication if both enhancer-targeted genes or promoter-targeted genes are the ones contributing to the transformation in these cells?

On page 13, line 281:

“FU-UR-1 proliferation was blunted by depletion of AT3, VCP, or both (Fig. 7a).”

What is the effect of the treatment with siRNA in other cells? Including in cells that do not have AT3 - important control.

On page 16, line 346

“For example, Hif1a was a strong AT3:VCP target in mouse ASPS tumors (Fig. 4e), but is not in FU-UR-1 cells, as previously noted.”

Is there is a difference in the CHIP seq in the different cells?

Line 348:

“We found similarly strong AT3:VCP peaks in ASPS-1 cells and human ASPS tumors around HIF1A.”

Similar in ASPS-1 and FU-UR-1? The wording is not clear

Page 17, line 350

“AT3.1/AT3.2 differences were noticeable in CB-5083 and C. manipulations of HEK293T cells”

Differences in what? The meaning is not clear

Reviewer #2 (Remarks to the Author):

ASPSCR1-TFE3 reprograms transcription by organizing enhancer 1 loops around hexameric VCP/p97 May 2023 Nature Communications

In this study Pozner, Verma, Li et al investigate the role of the fusion protein ASPSCR1-TF3 (AT3) in renal cell carcinoma (RCC) and alveolar soft part sarcoma (ASPS). To this end the authors use a combination of multiple biological models and orthogonal technologies (including proteomics and epigenetics) to identify a direct interaction between AT3 and the ATPase VCP, and delineate its functional role in these diseases. They also provide mechanistic insight about how this interaction participate in shaping the transcriptional and chromatin landscape of RCC and ASPS tumors, and the effect of its depletion or pharmacological inhibition on tumor growth in vitro and in vivo.

The study is nicely written and well structured, and most of the results supported by the experimental plan. However, this study has two major issues:

1) The use of two cell lines derived from completely different tumor types (RCC/FU-UR-1 and ASPS/ ASPS1) and expressing different AT3 isoforms are alternatively used to draw global conclusions. The lack of a direct comparison between the two cell line models through the entire experimental plan makes it extremely challenging to assess if the conclusions are well supported by the presented data. The authors must conduct a detailed comparison between the two lines (FU-UR-1 and ASPS1) at each step of the project, and provide clear data about the transcriptional, epigenetic and 3D structure similarity between these two models. They should also run a similar comparison with primary tumors to identify which fraction of these different profiles is also shared with them. Without this necessary comparison, any interpretation of the data and conclusions remains arbitrary.

2) The chromatin analyses presented in Figures 3, 4 and 6 are poorly described and analyzed. In particular there is no detailed QC about the quality of the data presented. How many replicates were performed for each ChIP-seq, Hi-ChIP and RNA-seq profiles? How well are the replicates correlating to each other, and how well do they correlate for the same

histone mark or AT3/VCP binding profile between the two models? More importantly, how was the AT3 ChIP-seq performed knowing that wt TF3 is indeed expressed in the same cells? All these points need to be addressed in great detail to be able to assess the reproducibility and robustness of the data.

Major points:

1. Figures 1A: why was the proteomic analysis conducted on the FU-UR-1 model and not on both models? How similar are the IP-ASPSCR1 proteomic profiles between the two tumor types? The authors should also comment about the expression of the wild-type isoform of ASPSCR1 in both models, and how this could impact the profile.
2. Figure 1B/C: given that the wild type TFE3 protein is also expressed in these models, how is this affecting the results shown in this figure? There seems to be significant differences between the two cell lines, the authors should address these points.
3. Figure 1E: it seems to be a higher interaction between VCP and AT3.2 compared to AT3.1, what is the functional implication for this difference in the biology of these two tumors, and their epigenetic profiles?
4. Figure 3: as discussed, a very detailed comparison should be provided between the chromatin profiles of the two models. The authors should mention how many replicates they have generated for each ChIP-seq track. They need to have at least two replicates for each, and show high correlation between them in the same model, as well as between models, to provide a robust rationale for the global function of AT3-VCP. The authors should also:
 - a. Provide pie charts illustrating the genome wide distribution of the ChIP-seq peaks for AT3 and VCP in each model.
 - b. Show the overlap between single AT3 peaks across the different model genome-wide by Venn diagrams, and not using stitched regions.

- c. Add the peaks numbers on each heatmap.
- d. Generate motif analyses for AT3 and VCP, for all the categories shown in Fig 3A.
- e. Explain how they generated the AT3 binding profiles knowing that wt AT3 is present and expressed in these cells.
- f. Provide evidence for the specificity of their AT3 peaks by running a similar ChIP-seq in both models upon AT3 or wt TFE3 depletion using the siRNAs they are showing.

5. Figure 4A: similar to the previous point, here the authors need to provide QC and replicates correlations for each histone mark in each model profiled. They also need to show similarity between the different models for each histone mark profiled using Venn diagrams. This should be separated into AT3 binding sites and other genomic regions to show if the similarity between models is higher at the fusion protein binding sites. They should also provide peak numbers for each heatmap. Finally, showing empty heatmaps for the K27me3 marks is not very useful. In case the authors do not detect any signal for this histone mark at these sites they should remove these heatmaps and just show the composite plots.

6. Figure 4G: here again the authors should provide a detailed comparison of the similarity in transcriptional changes upon siTFE3 KD between the two models, as well as between AT3 and VCP KD using Venn diagrams.

7. Figure 5A, B and C: what is the similarity between genome-wide transcriptional changes upon AT3/VCP KD in tumor lines and OE in HEK293T?

8. Figure 6: the entire figure suffers from a marked lack of QC and analyses. In particular:

- a. In Fig 6A the authors show pie charts illustrating the number of loops associated with K27ac signal in in FU-UR-1. It seems relatively unclear how they can identify only 12'000 loops, knowing that in multiple other normal and tumor models profiled using the same technology the number of loops is typically around 40'000. The authors once again need to specify the number of replicates and how they have selected and called the loops in greater

detail. In addition, they should specify in the figure which cell line was profiled.

b. The provided genome-wide analyses of changes in K27ac-associated Hi-ChIP loops is not informative. The authors should instead provide a scatter plot showing on each axis the log₂-normalized loop counts for the two conditions (siAT3 or VCP Vs siCTR) so the changes in each loop become clear to interpret.

9. The in vitro and in vivo results provided in Fig 7 and 8 are interesting, and provide a potential rationale for using VCP inhibitors as a therapeutic strategy in these diseases. However, VCP is ubiquitously expressed, and this gene is reported as an essential dependency in the Dependency Map Portal (Dep Map). Taken together, these observations point to a highly toxic effect of VCP pharmacological inhibition in vivo. If the authors want to substantiate their functional findings, they should show the effect of VCP KD and CB-5083 administration on a panel of normal cell types. Without these data, these observations lose their translational impact.

Minor points:

1. Figure 1H: to provide convincing data about their PLA results the authors should show higher magnification images and dot plots illustrating the number of spots detected in presence of both VCP and TFE3 antibodies, or only one of them with a IgG control.

2. Figure 4C: the authors should show the top 5 motifs detected at these sites with their corresponding p-values.

Reviewer #3 (Remarks to the Author):

In this study, Pozner et al. challenged to identify ASPSR1-TFE3 (AT3) collaborators that are important in alveolar soft part sarcoma (ASPS) as well as translocation-related renal cell carcinoma (tRCC). By immunoprecipitation followed by mass spectrometry they identified

VCP/p97. This result is cogent since previous studies by other groups clarified that VCP interacts with ASPSCR1 in cytoplasm. The authors then performed ChIP-seq and HiChIP to identify DNA co-binding of AT3 and VCP predominantly in enhancers and chromatin remodeling by the complex was suggested. AT3 and VCP interaction was important for maintenance and proliferation of ASPS and tRCC in vitro, and possible application of the therapy targeting VCP was discussed.

Experiments in the study is well designed and the results are convincing. The manuscript will attract interests of readers in the cancer research field. However, the manuscript also includes following points that preclude publication in its present status. I would request revisions to improve the manuscript.

Major points:

1. In the mass spectrometry-based proteomics, VCP is not a single candidate as AT3-associated proteins. It is a little hasty to conclude that VCP is the major AT3 collaborator in ASPS growth and transcriptional modulations. As a component of NuRD complex, MTA2 may also play an important role in cell survival and growth. Although the authors successfully exhibited upregulated expression of many target genes of the AT3/VCP axis such as GPNMB, SLC16A1 and ATG9B, they also demonstrated reverse effects in other genes such as MET, a known important AT3 target. I would speculate the potential role of MTA2 and the NuRD complex in MET suppression by AT3, and would request authors to address this question by performing MTA2 silencing.
2. In the previous study by the same group, a critical role of lactate metabolism in ASPS was reported. Although expression of SLC16A1 and SLC16A3 was mildly induced by co-expression of AT3 and VCP, modification of lactate metabolism was not indicated by AT3 and VCP interaction in the present study. The authors should clarify whether VCP actively regulates lactate import in ASPS by showing lactate content in the presence and absence of VCP.
3. Expression of the AT3 fusion protein induces predominant nuclear localization of VCP (Figure 1g), suggesting that the VCP function such as autophagy, protein ubiquitination and Golgi development may be impaired, and that this functional impairment may also be related to ASPS tumorigenesis. I would request authors to confirm whether normal VCP

function is affected by AT3.

4. Although TFE3 is invariably fused to ASPSCR1 in ASPS, it has various fusion partners in tRCC such as PRCC, SFPQ, or NONO. I wonder whether these fusions also interact with VCP and induce its nuclear recruitment. Simple experiments using other TFE3 fusions will support authors' conclusions that the modulation of VCP function is important in carcinogenesis.

5. It is important to demonstrate that the VCP inhibitor CB5083 suppresses VCP functions via AT3 interaction. Although the authors showed downregulation of AT3 target genes by CB5083 treatment, they did not show the direct relevance between inhibition of AT3/VCP-mediated chromatin remodeling and sarcoma growth, and it is possible that the growth suppression is non-specific toxicity of CB5083. Therefore, the authors should show IC50 of CB5083 in multiple cell lines including ASPS1, FU-UR-1, and non-ASPS tumor cells such as HCT116 and ASKA. Also, it is better to have results that exogenous expression of one or multiple target genes described in Figure 8h compensate the growth suppressive effect of CB5083 at least in part.

Minor points:

1. The canonical MiT/TFE binding motif is co-enriched for AT3, H3K27Ac and RNAPOL2 binding sites shown in Figure 4C. The authors should clarify whether the enrichment is same in the presence or absence of VCP.

2. AT3 enriched in three ASPS cells/groups are shown in Extended Fig. 3b. It is also better to have Venn diagrams that show overlapping of AT3 and VCP binding loci.

We are grateful to the Reviewers for the insightful comments provided in the critiques and the opportunity to address these by revision. Overall, we feel that we have strengthened the manuscript in our responses to Reviewers. The main themes of the critiques that we were able to answer with revisions were providing more of the quality control analyses of the genomics data and adding some controls to our testing of the transcriptional effects of VCP inhibition, using two cell lines that harbor other TFE3 fusions, which we confirmed do not interact with VCP. Although the criticism leveled that VCP is not a ready-for-clinic target for therapeutics is one with which we heartily agree, we also hope that the Reviewers will see the value in identifying any critical co-factor of a fusion oncoprotein, the likes of which have rarely been identified in cancer biology thus far. This is the first real investigation into the mechanism by which ASPSCR1::TFE3 achieves its profound transcriptomic reprogramming. Please find the Reviewer comments below in black text, followed by our responses in blue.

REVIEWER COMMENTS

Reviewer #1 (Remarks to the Author):

This is a solid study. There are a few points that would strengthen the article. In particular, it was not clear to this reader to what extent the inhibitions that were tried, whether by drug or siRNA, were specific to this tumor, do they affect other tumors, or do they affect all cells. For the sake of the reader, there are a few points throughout the manuscript that would benefit from clarification. For example:

What is a cofactor? In the introduction it was written: “We demonstrate that VCP is a likely obligate co-factor of ASPSCR1-TFE3, one of the only such fusion oncoprotein co-factors identified in cancer biology.”

When would you consider something a co-factor for an oncoprotein? For example, some variants of ApoE are important for allowing metastasis. Are they co-factors? I am not objecting to the use of the term, but it would help frame the questions you are addressing to define it better. For example, to be a co-factor does it have to engage directly with the oncoprotein? Since this could be a useful concept for other cancers, a clear definition would be helpful.

We thank the reviewer for this clarifying question. Although we did not know that we would identify a co-factor when we began this investigation, it makes perfect sense to define that term in the beginning of our presentation of our journey of discovery, since it has pertinence to where we landed. We have added the following text to the Introduction section of the paper:

“As a working definition, we defined a co-factor for a transcription factor fusion oncoprotein as a second protein that interacts directly with the oncoprotein, interacts at the site of function for that oncoprotein (at specific binding sites on chromatin), and enables the biological function of the fusion oncoprotein (The transcriptional impact of the fusion oncoprotein depends on the presence of the second protein.)”

What makes an oncoprotein “not readily targetable”

There are many reasons why different oncoproteins are not targetable. It might be useful here to list some of the different reasons and then, for those who do not work on AT3, what are the

specific reasons why AT3 not targetable? It would help the reader to determine what can be generalized from this work.

The literature focused on so-called “undruggable” transcription factors is too extensive to cover well in our paper. This reference below has been added, along with the explanatory text as follows:

“. . . given their large intrinsically disordered regions (IDRs) and lack of enzymatic sites for the binding of small molecules [Bushweller, 2019].”

On page 3, line 74 it is written: “Here, we identify VCP as the most important nuclear interactor with the fusion oncoprotein AT3.”

It would help to say, What are the criteria for saying it the most important. Does it bind the most? Please clarify.

We have adjusted this text to reflect our statement of co-factor definition earlier in the Introduction:

“Here, we identified VCP as a co-factor to AT3, beginning with. . .”

On page 7, lines 130-133:

“Co-transfection of CΔ with AT3 reduced the presence of higher molecular weight assemblies in FLAG-AT3-IP (Fig. 2g). FLAG-AT3-IP without co-transfected CΔ recovered assemblies as large as 1050 kD, consistent with a VCP hexamer complexed with six AT3 molecules.”

Did the transfection of CΔ disassemble the AT3 in the nucleus?

These are all excellent questions. We asked each of them ourselves during our investigation. We demonstrated that CΔ competes with AT3 to disassemble VCP hexamers, in vitro, in **Fig. 2f** and **Fig. 2h**, as well as in the associated Extended Data Figure, panel **e**. The loss of the higher molecular weight assemblies including AT3 and VCP as demonstrated in **Fig. 2g** demonstrates the impact when these biologicals of VCP hexamer assembly stabilized by AT3 and hexamer disassembly mediated by CΔ compete in cells co-transfected with both. We interpret the loss of the higher molecular weight assemblies as evidence that CΔ disassembled the VCP hexamers with 6 AT3 molecules at least partly.

The challenge with our text, which you quoted, is that we are trying to be as plainly descriptive of the data as possible in this Results section, without interpretation. The problem this creates is that the reader has to guess what we are thinking, which is difficult in so dense a paper. We have adjusted that text with the addition below:

“The loss of the higher molecular weight assemblies with co-transfected CΔ, indicates that it at least dynamically destabilizes what are otherwise AT3:VCP double hexamers.”

Did CΔ need a nuclear localization signal to have an effect?

While the nuclear localization signal may or may not have been necessary for such a small protein to enter the nucleus, our goal was to localize the VCP disassembly effect as much to the nucleus as possible, given the difficulty otherwise with differentiating nuclear VCP impacts from cytoplasmic VCP impacts, the latter of which are AT3-independent. Attempts to transfect CΔ-NLS into FU-UR-1 and ASPS-1 cells had such profound physiological effects that the cells were

not viable, leading us to omit these experiments from this particular manuscript, until we can identify the means of conducting a well-controlled experiment to demonstrate this phenomenon.

Did it affect the cell physiology?

The HEK293T cell model is not ideal for assessing cell physiology types of questions, as even at baseline, this is a rapidly proliferating cell line. However, the most important physiological impact of Δ co-transfection with AT3 in HEK was observed very carefully in transcription assays by RNA-seq in **Fig. 5d**.

On page 12, line 265-266:

“AT3 depletion correlated better with VCP depletion for transcriptional changes among enhancer-targeted genes than among promoter-targeted genes.”

Is there any indication if both enhancer-targeted genes or promoter-targeted genes are the ones contributing to the transformation in these cells?

This is an excellent question. We intend to pursue this aggressively in future work. Unfortunately, aside from interrogating large series of specific target genes either individually or through large genomic screens, we cannot cleanly separate the two. This paper already aggressively pushes limits on the content for a single paper. We will therefore save these other investigations for future work. We have noticed that VCP depletion and VCP enzymatic inhibition differ in their impacts on proliferation (**Fig. 7d**), but even this is tantamount to comparing apples to oranges as the penetrance of each method (siRNA, small molecule inhibitor) is not equivalent to the other.

On page 13, line 281:

“FU-UR-1 proliferation was blunted by depletion of AT3, VCP, or both (Fig. 7a).”

What is the effect of the treatment with siRNA in other cells? Including in cells that do not have AT3 - important control.

We agree that the impact of VCP depletion is important to test in the presence and absence of AT3, which is why the two were combined for these assays. Importantly, VCP depletion had no additive or synergistic effect over AT3 depletion (**Fig. 7a**). The importance of application of TFE3-directed siRNAs to other cells, which lack AT3 and usually lack TFE3 expression itself would provide a rather not interesting control, in our opinion. TFE3 is not active in the nucleus of most cells, unless they are in extreme metabolic stress. The challenge with VCP depletion in other cells is that it will usually blunt their proliferation, but through a completely different mechanism, because VCP is such an important protein in the cytoplasm—and in some cases in the nucleus—of almost all cells. Critically, it does not have the same effect on transcription in other cells, as evidenced by **Extended Data Fig. 6i** that has been added to this revision.

On page 16, line 346

“For example, Hif1a was a strong AT3:VCP target in mouse ASPS tumors (Fig. 4e), but is not in FU-UR-1 cells, as previously noted.”

Is there is a difference in the ChIP seq in the different cells?

The depicted data in **Fig. 4e** are ChIP-seq enrichment, precisely. The same peaks are not present in FU-UR-1 AT3 or VCP ChIP-seq, nor have been in previous assessments of FU-UR-1

for AT3 binding sites. We have reworded this per the text noted below in the following Critique item.

Line 348:

“We found similarly strong AT3:VCP peaks in ASPS-1 cells and human ASPS tumors around HIF1A.”

Similar in ASPS-1 and FU-UR-1? The wording is not clear

We have reworded these two statements as follows:

“There are highly enriched AT3:VCP ChIP-seq peaks around the *Hif1a* locus in mouse ASPS tumors (**Fig. 4e**) and around the homologous *HIF1A* locus in ASPS-1 cells and human ASPS tumors. No AT3:VCP peaks were identified in FU-UR-1 cells near *HIF1A*, as previously reported [Kobos 2013].”

Page 17, line 350

“AT3.1/AT3.2 differences were noticeable in CB-5083 and CΔ manipulations of HEK293T cells”
Differences in what? The meaning is not clear

We struggled to fit detailed explanations into this very circumscribed text length. We have reworded this as follows:

“In the transcription assays in HEK293T cells (**Fig. 5d and 5h**), the impact of CB-5083 and CΔ were noticeably different between AT3.1 and AT3.2, the effects being more pronounced with AT3.1.”

Reviewer #2 (Remarks to the Author):

ASPSCR1-TFE3 reprograms transcription by organizing enhancer 1 loops around hexameric VCP/p97 May 2023 Nature Communications

In this study Pozner, Verma, Li et al investigate the role of the fusion protein ASPSCR1-TF3 (AT3) in renal cell carcinoma (RCC) and alveolar soft part sarcoma (ASPS). To this end the authors use a combination of multiple biological models and orthogonal technologies (including proteomics and epigenetics) to identify a direct interaction between AT3 and the ATPase VCP, and delineate its functional role in these diseases. They also provide mechanistic insight about how this interaction participate in shaping the transcriptional and chromatin landscape of RCC and ASPS tumors, and the effect of its depletion or pharmacological inhibition on tumor growth in vitro and in vivo.

The study is nicely written and well structured, and most of the results supported by the

experimental plan. However, this study has two major issues:

1) The use of two cell lines derived from completely different tumor types (RCC/FU-UR-1 and ASPS/ ASPS1) and expressing different AT3 isoforms are alternatively used to draw global conclusions. The lack of a direct comparison between the two cell line models through the entire experimental plan makes it extremely challenging to assess if the conclusions are well supported by the presented data. The authors must conduct a detailed comparison between the two lines (FU-UR-1 and ASPS1) at each step of the project, and provide clear data about the transcriptional, epigenetic and 3D structure similarity between these two models. They should also run a similar comparison with primary tumors to identify which fraction of these different profiles is also shared with them. Without this necessary comparison, any interpretation of the data and conclusions remains arbitrary.

We fully agree that this is an important step. This was the impetus behind our extensive analyses in what are now **Extended Data Fig. 4a-b, Fig. 4, and Extended Data Fig. 5**, demonstrating the striking similarities between even the different species and certainly the different human cell lines. We have added **Extended Data Figures 3 and 6a-c**, per the specific requests below, as well.

There have been other investigations into the genome-wide chromatin distribution of AT3 as well as into its target genes. We therefore did not emphasize our findings in this space as much as those prior studies, which our findings generally corroborated. As is often the case, most such studies in rare tumor types like ASPS take significant liberties with detailed comparisons between every cell line and most forego human tumor comparisons altogether, except for a few pointed analyses. We did not arbitrarily select different cell lines for different analyses. We performed most of our analyses in FU-UR-1 cells, which are the best growing and most manipulable of the natively AT3-expressing cancer cells. The fact that we corroborated the findings of many of our experiments in mouse whole tumor specimens and even human tumor specimens as well as a second cell line is almost unprecedented for this rare of a tumor type.

2) The chromatin analyses presented in Figures 3, 4 and 6 are poorly described and analyzed. In particular there is no detailed QC about the quality of the data presented. How many replicates were performed for each ChIP-seq, Hi-ChIP and RNA-seq profiles?

The replicates for every ChIP-seq, HiChIP and RNA-seq profile are delineated in each figure legend in **Fig. 3a, 4d, 6a-d, Extended Data Fig. 5a-d**.

How well are the replicates correlating to each other, and how well do they correlate for the same histone mark or AT3/VCP binding profile between the two models?

We have added correlation heatmaps for each replicate that did not already have correlations shown, now as **Extended Data Fig. 3a-d and 5a**. Not every chromatin mark was analyzed in every model. Most of the chromatin marks were analyzed only in human tumors and mouse tumors, whereas H3K27ac was also analyzed in human FU-UR-1 cells. AT3 and VCP, as well as RNAPOL2 were assessed by ChIP-seq in ASPS-1, FU-UR-1, human ASPS tumors and mouse ASPS tumors, as this addressed the central investigational question of VCP and AT3 co-localization across the genome.

More importantly, how was the AT3 ChIP-seq performed knowing that wt TF3 is indeed expressed in the same cells?

Wildtype TFE3 is not expressed in FU-UR-1 cells. FU-UR-1 is a male cell line and *TFE3* is on the X chromosome, meaning that the only copy of *TFE3* in the genome of these cells is split by involvement in the chromosomal translocation. That said, we did not use anti-TFE3 antibodies for ChIP-seq for any experiments in this paper. We used anti-ASPSCR1 antibodies for ChIP-seq, as indicated previously in the methods resources table, but now also added as text to the Results section:

“ . . . for AT3 ChIP-seq (using an antibody against the amino terminus of ASPSCR1). . . ”

Native ASPSCR1 is expressed in all these cells in parallel to AT3 expression, but native ASPSCR1 is exclusively cytoplasmic in location. We confirmed this known protein distribution by the observation of only a single band by ASPSCR1 western blot (at the size of AT3 and co-localized by TFE3 western blot) after nuclear isolation in our first experiments performing IP-MS with the same anti-ASPSCR1 antibody. Some of these blots are included below in this Response to Reviewers document.

All these points need to be addressed in great detail to be able to assess the reproducibility and robustness of the data.

We found it a significant challenge to include all of these details, given the imposed constraints to manuscript length. We have attempted to strengthen our Extended Data to provide the requested details, in addition to pointing out the details that were already included in the original submission, but overlooked due to the overall density of the paper.

Major points:

1. Figures 1A: why was the proteomic analysis conducted on the FU-UR-1 model and not on both models?

We conducted the original proteomic analysis on the only cell line we had at the time as well as on mouse whole tumor specimens, for optimal spread between the contexts for AT3 expression between the two experimental models. One was a cell line, the other a whole tumor. One was an RCC, the other a sarcoma. One was human, the other mouse. We perform such comparative cross-species experiments based on the premise that the most important (with regard to oncogenesis) interactions or gene targets or chromatin impacts of any fusion oncoprotein will be those that are noticeably shared between tumors initiated by expression of the fusion oncoprotein in two related, but distinct species, mouse and human. This may be a premise against which many very cogent arguments can be made, but it is simply what we do as mouse modelers. We later corroborated the IP results by western blot in the other cell line, but have not performed the additional mass spectroscopy proteomics analysis for ASPS-1 cells. As interested as we are in the ASPS-1 cell line AT3 interactome, those data did not seem necessary for the trajectory of this specific report, which primarily focused on a single interactor, VCP.

How similar are the IP-ASPSCR1 proteomic profiles between the two tumor types?

As demonstrated in **Fig. 1a** and **Extended Data Figure 1a**, the nuclear AT3 interactome is extremely similar between these two most disparate contexts of human RCC cells and mouse ASPS tumors.

The authors should also comment about the expression of the wild-type isoform of ASPSCR1 in both models, and how this could impact the profile.

Wild-type ASPSCR1 is present exclusively in the cytoplasm. Below are anti-ASPSCR1 western blots from the IP-ASPSCR1 samples after nuclear extract. We are happy to include these as an Extended Data Figure, if requested. Native ASPSCR1 would run at 61kD, if it were present in these nuclei.

2. Figure 1B/C: given that the wild type TFE3 protein is also expressed in these models, how is this affecting the results shown in this figure?

The reason why we performed the experiment using anti-ASPSCR1 antibody, anti-TFE3 antibody, and anti-Interactor antibody for each interaction was to triangulate around this issue. There is no presence of wild-type TFE3 on the anti-Interactor antibody IPs for any of these, likely because there is so little wild-type TFE3 present in the nucleus of these cells, as well as most cells lacking Xp11 rearrangements. We have now included IP:WBs for nuclear TFE3 in other cell lines that lack ASPSCR1 as a fusion partner, instead bearing PRCC-TFE3 and NONO-TFE3. These are now included as **Extended Data Fig. 6h**.

There seems to be significant differences between the two cell lines, the authors should address these points.

Each of these IP-WBs was performed as a separate experiment, with internal controls. Interpreting comparisons between them beyond the observation that the interaction was demonstrable in comparison to IgG (negative) and input (positive) controls is not permitted by this type of experiment.

3. Figure 1E: it seems to be a higher interaction between VCP and AT3.2 compared to AT3.1, what is the functional implication for this difference in the biology of these two tumors, and their epigenetic profiles?

Noticing this is strong testament to the Reviewer's perspicacity. The real biology behind this discrepancy was actually that this experiment was not controlled for the expression level of any of these IP baits, and in that particular run, AT3.2 was more highly expressed than AT3.1. We have noted this in the legend to warn readers from over-interpretation, but have not repeated the experiment for a better blot, as it is not critical to the question asked by this experiment,

which was really whether or not there were interactions with the interrogated portions of the fusion oncoprotein. For careful comparisons between AT3.1 and AT3.2, please see **Fig. 5**, where expression levels were more carefully balanced.

4. Figure 3: as discussed, a very detailed comparison should be provided between the chromatin profiles of the two models. The authors should mention how many replicates they have generated for each ChIP-seq track. They need to have at least two replicates for each, and show high correlation between them in the same model, as well as between models, to provide a robust rationale for the global function of AT3-VCP.

We included at least 2 replicates for each ChIP-seq track. We have added correlation heatmaps as the new **Extended Data Fig. 3a-d**.

The authors should also:

a. Provide pie charts illustrating the genome wide distribution of the ChIP-seq peaks for AT3 and VCP in each model.

Please see the new **Extended Data Fig. 3g**.

b. Show the overlap between single AT3 peaks across the different model genome-wide by Venn diagrams, and not using stitched regions.

Please see the new **Extended Data Fig. 3f**.

c. Add the peaks numbers on each heatmap.

These were already depicted by noted heatmap height criteria in **Fig. 3a** and by the associated histogram plots and numbers listed in **Fig. 4a** and **Extended Data Fig. 5a-d**. The specific numbers are also now noted in the requested Venn diagrams of new **Extended Data Fig. 3e**.

d. Generate motif analyses for AT3 and VCP, for all the categories shown in Fig 3A.

Please see **Extended Data Fig. 3h**.

e. Explain how they generated the AT3 binding profiles knowing that wt AT3 is present and expressed in these cells.

This comment was probably mistyped. There is no such entity as wt AT3, which is an acronym for ASPSCR1::TFE3. If this refers to wt TFE3, the comment is immaterial, as the antibody used for ChIP-seq was anti-ASPSCR1, as explained above. If this refers to wt ASPSCR1, this is also not a concern as native ASPSCR1 is a cytoplasmic protein with no nuclear presence, as indicated above, where westerns show only a band at the AT3 size after nuclear extract IP with the anti-ASPSCR1 antibody.

f. Provide evidence for the specificity of their AT3 peaks by running a similar ChIP-seq in both models upon AT3 or wt TFE3 depletion using the siRNAs they are showing.

It is possible that this item was requested due to the misunderstanding that we were using an anti-TFE3 antibody for ChIP-seq, which we were not. However, a negative (usually only partly) ChIP-seq for a protein after its depletion has been performed for a few first-time ChIP analyses for a few proteins. We have not performed this, as we prefer the ENCODE standard of using input chromatin as a negative control, measuring enrichment of the ChIP-seq specimen over this input control. As a biological positive control for AT3 ChIP-seq, we have performed the ChIP-seq in two human cell lines from different types of cancer that share only the fusion oncoprotein expression, as well as human whole tumor specimens and mouse whole tumor specimens, all of which show striking similarities in the distribution of AT3 across their respective genomes.

5. Figure 4A: similar to the previous point, here the authors need to provide QC and replicates correlations for each histone mark in each model profiled.

The replicates correlations have been added to **Extended Data Fig. 5a**.

They also need to show similarity between the different models for each histone mark profiled using Venn diagrams. This should be separated into AT3 binding sites and other genomic regions to show if the similarity between models is higher at the fusion protein binding sites.

This is a very interesting idea. We are grateful for the suggestion. The similarity between the models is much more pronounced at AT3 binding sites, as expected. The requested Venn diagrams are provided in **Extended Data Fig. 5e**.

They should also provide peak numbers for each heatmap.

The numbers of peaks in each heatmap are represented by the relative height of each heatmap in **Fig. 3a**, as indicated by the scale bar to the lower left and by the number of peaks listed of each type in **Fig. 4a** and **Extended Data Fig. 4b-d**. While this requires readers to add intergenic, exonic, and intronic peaks to identify the numbers of distal peaks, it seems more efficient with space to list these numbers once, rather than multiple times. We have added the peak numbers for each heatmap to the figure legends.

Finally, showing empty heatmaps for the K27me3 marks is not very useful. In case the authors do not detect any signal for this histone mark at these sites they should remove these heatmaps and just show the composite plots.

Perhaps the resolution of the images available to the reviewers was not optimal, but these are not completely empty heatmaps. The scale for the H3K27me3 heatmaps has been quartered to show a bit more of the scant signal that is there in the revised **Fig. 4a**.

6. Figure 4G: here again the authors should provide a detailed comparison of the similarity in transcriptional changes upon siTFE3 KD between the two models, as well as between AT3 and VCP KD using Venn diagrams.

These comparisons have been added as **Extended Data Fig. 6a-b**.

7. Figure 5A, B and C: what is the similarity between genome-wide transcriptional changes upon AT3/VCP KD in tumor lines and OE in HEK293T?

Venn diagram comparison between AT3 knock-down in ASPS-1 cells and AT3 expression in HEK293T cells has been added as **Extended Data Fig. 6c**.

8. Figure 6: the entire figure suffers from a marked lack of QC and analyses. In particular:

a. In Fig 6A the authors show pie charts illustrating the number of loops associated with K27ac signal in in FU-UR-1. It seems relatively unclear how they can identify only 12'000 loops, knowing that in multiple other normal and tumor models profiled using the same technology the number of loops is typically around 40'000. The authors once again need to specify the number of replicates and how they have selected and called the loops in greater detail. In addition, they should specify in the figure which cell line was profiled.

The details of loop calling and filtering are explained in the Methods section, page 34-5. It is true that these Hi-ChIP experiments in the FU-UR-1 cell line were not astonishingly robust, but they did render very confident loops for comparison. We have since added Hi-ChIP profiling of the ASPS-1 cell line, which rendered a more typical number of loops, using the same stringency parameters. We have added data from this analysis and a comparison between the two cell lines as **Fig. 6a** and **Extended Data Fig. 7a-e**.

b. The provided genome-wide analyses of changes in K27ac-associated Hi-ChIP loops is not informative. They authors should instead provide a scatter plot showing on each axis the log₂-normalized loop counts for the two conditions (siAT3 or VCP Vs siCTR) so the changes in each loop become clear to interpret.

We regret that the reviewer sees no value in the changes in H3K27ac ChIP-seq enrichment. Such data are not standard and were very difficult to generate, but we understand that the demonstration of absent H3K27ac-Hi-ChIP-defined loops in a second condition leads to the necessary question of what happened to cause this deficit in loops. Is it the absence of H3K27ac or a true shift in chromatin conformation? Our rigorous experiments to address H3K27ac enrichment as directly reduced from depletion of AT3 or VCP argue that what is changing is at least the presence of H3K27ac histone marks, a surrogate of the “enhancer-ness” of those loci.

We have added the requested scatter plot for the conditions for which we have Hi-ChIP data as **Fig. 6g**.

9. The in vitro and in vivo results provided in Fig 7 and 8 are interesting, and provide a potential rationale for using VCP inhibitors as a therapeutic strategy in these diseases. However, VCP is ubiquitously expressed, and this gene is reported as an essential dependency in the Dependency Map Portal (Dep Map). Taken together, these observations point to a highly toxic effect of VCP pharmacological inhibition in vivo. If the authors want to substantiate their functional findings, they should show the effect of VCP KD and CB-5083 administration on a panel of normal cell types. Without these data, these observations lose their translational impact.

As expressed in the Discussion section, we doubt the translational impact of CB-5083 from our own data. Rather than test toxicity of the drug in a panel of normal cell lines, we applied a more rigorous test of toxicity: the application of drug to a living host model system. While this worked to blunt tumor growth, it was not a sufficiently strong effect to merit clinical development of CB-

5083 as a drug for ASPS. The problem was the very toxicity that one would expect, even if that toxicity is not entirely on-target, per the many other reports on CB-5083.

We have added experiments to demonstrate the effect of VCP inhibition on other cells with similar TFE3-orchestrated transcriptomes, but no VCP interaction with their particular fusion oncoproteins. CB-5083 in these cells importantly has the opposite effect on transcription of AT3 target genes, even if it is toxic to the cells at similar concentrations. Our point is that VCP is an important biological co-factor of the fusion, with structural data available that makes pharmacological manipulation possible, not necessarily that the chemical matter currently available is going to be answer for these types of cancer, yet.

Minor points:

1. Figure 1H: to provide convincing data about their PLA results the authors should show higher magnification images and dot plots illustrating the number of spots detected in presence of both VCP and TFE3 antibodies, or only one of them with a IgG control.

Although the explicit number of dots was not discretely measurable, the software was able to quantify the signal intensity of the PLA signal per nucleus. We have added this requested quantitative data as **Fig. 1i**. We have added the requested higher magnification images as **Extended Data Fig. 1g**.

2. Figure 4C: the authors should show the top 5 motifs detected at these sites with their corresponding p-values.

We have added the requested data as **Extended Data Figure 3h**.

Reviewer #3 (Remarks to the Author):

In this study, Pozner et al. challenged to identify ASPSR1-TFE3 (AT3) collaborators that are important in alveolar soft part sarcoma (ASPS) as well as translocation-related renal cell carcinoma (tRCC). By immunoprecipitation followed by mass spectrometry they identified VCP/p97. This result is cogent since previous studies by other groups clarified that VCP interacts with ASPSCR1 in cytoplasm. The authors then performed ChIP-seq and HiChIP to identify DNA co-binding of AT3 and VCP predominantly in enhancers and chromatin remodeling by the complex was suggested. AT3 and VCP interaction was important for maintenance and proliferation of ASPS and tRCC in vitro, and possible application of the therapy targeting VCP was discussed.

Experiments in the study is well designed and the results are convincing. The manuscript will attract interests of readers in the cancer research field. However, the manuscript also includes following points that preclude publication in its present status. I would request revisions to improve the manuscript.

Major points:

1. In the mass spectrometry-based proteomics, VCP is not a single candidate as AT3-associated proteins. It is a little hasty to conclude that VCP is the major AT3 collaborator in ASPS growth and transcriptional modulations. As a component of NuRD complex, MTA2 may also play an important role in cell survival and growth. Although the authors successfully exhibited upregulated expression of many target genes of the AT3/VCP axis such as GPNMB, SLC16A1 and ATG9B, they also demonstrated reverse effects in other genes such as MET, a known important AT3 target. I would speculate the potential role of MTA2 and the NuRD complex in MET suppression by AT3, and would request authors to address this question by performing MTA2 silencing.

We completely agree that MTA2 and the NuRD complex are prime candidates for investigation as interactors with AT3. Such studies are underway in our laboratory. This paper focused—perhaps arbitrarily—on VCP as an interactor with AT3 at both upregulated and downregulated targets, due to the fact that VCP was the only interactor that was specific to the ASPSCR1 portion of AT3 (**Fig. 1e**), whereas MTA2 and MATR3 interacted with the TFE3 portion of AT3. That does not mean that NuRD is not interesting, but we certainly hope that there is more than one publication available to report the transcriptional regulation mechanisms of AT3. We have already crammed 130 experiments and 500GB of sequencing data into this singular manuscript, and found it very difficult to fit the explanations necessary for their clear presentation, given the text length constraints. We therefore beg to be permitted to save the investigation of NuRD interactions with AT3 for future manuscripts. While NuRD is of interest, details of its interaction will not invalidate any of the findings related to AT3:VCP.

2. In the previous study by the same group, a critical role of lactate metabolism in ASPS was reported. Although expression of SLC16A1 and SLC16A3 was mildly induced by co-expression of AT3 and VCP, modification of lactate metabolism was not indicated by AT3 and VCP interaction in the present study. The authors should clarify whether VCP actively regulates lactate import in ASPS by showing lactate content in the presence and absence of VCP.

Lactate metabolism related to AT3 remains a vibrant interest of our laboratory, as well. It was simply not the focus of this series of investigations, which focused on the transcriptional impact of VCP as a co-factor for AT3, rather than the downstream targeted biology from that transcriptional impact.

3. Expression of the AT3 fusion protein induces predominant nuclear localization of VCP (Figure 1g), suggesting that the VCP function such as autophagy, protein ubiquitination and Golgi development may be impaired, and that this functional impairment may also be related to ASPS tumorigenesis. I would request authors to confirm whether normal VCP function is affected by AT3.

We completely agree that the metabolic impacts of partial VCP re-localization to the nucleus are extremely interesting and worthy of additional experiments, which we are pursuing. We could not fit them into this primary investigation, which focused on the AT3:VCP interaction on chromatin. At baseline, we must acknowledge that VCP is one of the most abundantly expressed proteins in most cells, making its relative depletion in any compartment somewhat unlikely. We also must carefully acknowledge that we do not see an absence of VCP in the cytoplasm in AT3 expressing cells, generally (**Fig. 1f** and **Extended Data Fig. 1c-d**). The

strength of VCP presence in the nucleus is noteworthy in the presence of AT3 expression, but it does not become as exclusively nuclear as AT3 itself. The re-localization of VCP from the cytoplasm to the nucleus by AT3 expression in HEK293T cells is a powerful example of the strength of their interaction, but also reflects the somewhat rare circumstance of cytoplasmic VCP exclusivity in HEK293T cells at baseline. Most other cells have some VCP in the nucleus at baseline, even without AT3. We hope not to mislead readers into thinking that any nuclear VCP is evidence of AT3 expression. We tried to explain this carefully in the first lines at the top of page 6. Overall, this means that impaired cytoplasmic VCP function is likely to be subtle overall and rather difficult to tease out in experiments, making it better for the focus of another series of investigations, reported separately.

4. Although TFE3 is invariably fused to ASPSCR1 in ASPS, it has various fusion partners in tRCC such as PRCC, SFPQ, or NONO. I wonder whether these fusions also interact with VCP and induce its nuclear recruitment. Simple experiments using other TFE3 fusions will support authors' conclusions that the modulation of VCP function is important in carcinogenesis.

We have added TFE3 IPs in PRCC-TFE3 and NONO-TFE3 expressing cell lines, showing no co-IP of VCP with these other fusion oncoproteins as **Extended Data Figure 6h**.

5. It is important to demonstrate that the VCP inhibitor CB5083 suppresses VCP functions via AT3 interaction. Although the authors showed downregulation of AT3 target genes by CB5083 treatment, they did not show the direct relevance between inhibition of AT3/VCP-mediated chromatin remodeling and sarcoma growth, and it is possible that the growth suppression is non-specific toxicity of CB5083. Therefore, the authors should show IC50 of CB5083 in multiple cell lines including ASPS1, FU-UR-1, and non-ASPS tumor cells such as HCT116 and ASKA. Also, it is better to have results that exogenous expression of one or multiple target genes described in Figure 8h compensate the growth suppressive effect of CB5083 at least in part.

The non-AT3-related effects of CB-5083 are substantial, and make VCP a somewhat difficult-to-tease-out target of drug inhibition in AT3-related cancers. Indeed, most cells are inhibited at some level by VCP inhibition, largely due to disruption of its cytoplasmic roles in protein folding quality control and autophagy. We propose that this non-specificity of VCP as a dependency does not make the elucidation of its biology as an AT3 co-factor any less important, even if it makes it more problematic as a drug target in ASPS. We have added experiments using cell lines with related transcriptomes by TFE3 targeting, demonstrating that most AT3 target genes have the opposite changes in transcription by application of CB-5083 (**Extended Data Figure 6i**). Although this only addresses the challenge partly, it highlights the biology of VCP in complex with AT3.

Minor points:

1. The canonical MiT/TFE binding motif is co-enriched for AT3, H3K27Ac and RNAPOL2 binding sites shown in Figure 4C. The authors should clarify whether the enrichment is same in the presence or absence of VCP.

We have added the requested motif analyses in **Extended Data Figure 3h**.

2. AT3 enriched in three ASPS cells/groups are shown in Extended Fig. 3b. It is also better to have Venn diagrams that show overlapping of AT3 and VCP binding loci.

We have added the requested Venn diagrams in **Extended Data Figure 3e**.

REVIEWERS' COMMENTS

Reviewer #1 (Remarks to the Author):

I have read the revised manuscript, my review and the responses, and the other two reviews and the responses. I still think, as I did before, that it is a very solid manuscript with interesting and important data. The responses to many of my, and the other questions were thoughtful. As the authors pointed out they could have continued to address new questions, but they were pushing the limits of what would fit. I think that the work is solid enough that it is time to share it with the community because it will inform work done by others.

Reviewer #2 (Remarks to the Author):

The authors have convincingly replied to all my comments.

I would also like to apologize for misinterpreting the experimental conditions used to perform the AT3 ChIP-seq profiling. Given the cytoplasmic location of wt ASPSCR1, using an antibody against the amino terminus of ASPSCR1 to profile AT3 chromatin occupancy makes sense.

Reviewer #3 (Remarks to the Author):

My comments at the first review have been reasonably addressed by the authors. I do not have any additional comments.